# Cell cycle transcriptomics of *Capsaspora* provides insights into the evolution of cyclin-CDK machinery

**Alberto Pérez-Posada**[1], **Omaya Dudin**[1], **Eduard Ocaña-Pallarès**[1], **Iñaki Ruiz-Trillo**[1,2,3]*, **Andrej Ondracka**[1]*

**1** Institut de Biologia Evolutiva (CSIC-Universitat Pompeu Fabra), Barcelona, Catalonia, Spain, **2** Departament de Genètica, Microbiologia i Estadística, Universitat de Barcelona, Barcelona, Catalonia, Spain, **3** ICREA, Barcelona, Catalonia, Spain

* inaki.ruiz@ibe.upf-csic.es (IRT); andrej.ondracka@gmail.com (AO)

## Abstract

Progression through the cell cycle in eukaryotes is regulated on multiple levels. The main driver of the cell cycle progression is the periodic activity of cyclin-dependent kinase (CDK) complexes. In parallel, transcription during the cell cycle is regulated by a transcriptional program that ensures the just-in-time gene expression. Many core cell cycle regulators are widely conserved in eukaryotes, among them cyclins and CDKs; however, periodic transcriptional programs are divergent between distantly related species. In addition, many otherwise conserved cell cycle regulators have been lost and independently evolved in yeast, a widely used model organism for cell cycle research. For a better understanding of the evolution of the cell cycle regulation in opisthokonts, we investigated the transcriptional program during the cell cycle of the filasterean *Capsaspora owczarzaki*, a unicellular species closely related to animals. We developed a protocol for cell cycle synchronization in *Capsaspora* cultures and assessed gene expression over time across the entire cell cycle. We identified a set of 801 periodic genes that grouped into five clusters of expression over time. Comparison with datasets from other eukaryotes revealed that the periodic transcriptional program of *Capsaspora* is most similar to that of animal cells. We found that orthologues of cyclin A, B and E are expressed at the same cell cycle stages as in human cells and in the same temporal order. However, in contrast to human cells where these cyclins interact with multiple CDKs, *Capsaspora* cyclins likely interact with a single ancestral CDK1-3. Thus, the *Capsaspora* cyclin-CDK system could represent an intermediate state in the evolution of animal-like cyclin-CDK regulation. Overall, our results demonstrate that *Capsaspora* could be a useful unicellular model system for animal cell cycle regulation.

## Author summary

When cells reproduce, proper duplication and splitting of the genetic material is ensured by cell cycle control systems. Many of the regulators in these systems are present across all eukaryotes, such as cyclin and cyclin-dependent kinases (CDK), or the E2F-Rb

**Data Availability Statement:** All relevant data are within the manuscript and its Supporting Information files. RNAseq reads are deposited in the Gene Expression Omnibus (GEO) under the

accession number GSE141156 and will be available after acceptance. The Supporting Information files are also available from figshare: https://figshare.com/s/4d642c9854efe6d879a7.

**Funding:** This work was funded by a European Research Council Consolidator Grant (ERC-2012-Co-616960) to IRT; and a grant from the Spanish Ministry for Economy and Competitiveness (MINECO; BFU2017-90114-P, with European Regional Development Fund support) to IRT. AP was supported by a "la Caixa" Foundation (ID 100010434) fellowship, whose code is LCF/BQ/ES16/11570008. OD was supported by a Swiss National Science Foundation Early PostDoc Mobility fellowship (P2LAP3_171815) and a Marie Sklodowska-Curie individual fellowship (MSCA-IF 746044). EOP was supported by a pre-doctoral FPI grant from MINECO. AO was supported by a Marie Sklodowska-Curie individual fellowship (MSCA-IF 747086). The funders had no role in study design, data collection and analysis, decision to publish, or preparation of the manuscript.

**Competing interests:** No authors have declared competing interests.

transcriptional network. Opisthokonts, the group comprising animals, fungi and their unicellular relatives, represent a puzzling scenario: in contrast to animals, where the cell cycle core machinery seems to be conserved, studies in yeasts have shown that some of these regulators have been lost and independently evolved. For a better understanding of the evolution of the cell cycle regulation in opisthokonts, and ultimately in the lineage leading to animals, we have studied cell cycle regulation in *Capsaspora* owczarzaki, a unicellular amoeba more closely related to animals than fungi. Our findings suggest that, in the ancestor of *Capsaspora* and animals, cyclins oscillate in the same temporal order as in animals, and that expansion of CDKs occurred later in the lineage that led to animals.

## Introduction

The cell cycle is an essential and fundamental biological process that underpins the cell division and proliferation of all cells. Progression through the cell cycle involves multiple layers of regulation [1]. The main regulatory networks that govern the transitions between cell cycle stages are broadly conserved in eukaryotes, both on the level of individual regulators [2], as well as on the level of network topology [3]. However, it is still not well understood how this conserved regulatory network is deployed in cells with different cellular lifestyles and how it changes across evolution.

Among the main regulators of the progression through the cell cycle are cyclins and cyclin-dependent kinases (CDKs), two gene families broadly conserved across eukaryotes [1,2]. Cyclins and CDKs have undergone independent expansions and subfunctionalization in every major lineage of eukaryotes, including opisthokonts [4–6]. In animals, there are multiple cyclins and CDKs that form discrete complexes, activating specific downstream effectors in different phases of the cell cycle [7,8]. Cyclin D-CDK4,6 complexes control entry into the cell cycle in response to mitogenic factors [9–11]. The G1/S transition is driven by the Cyclin E/CDK2 complex [12,13], and progression through S phase is controlled by the Cyclin A/CDK2 complex [13]. Lastly, cyclin B/CDK1 drive completion of mitosis [14,15]. In contrast, in the budding yeast *Saccharomyces cerevisiae* one single CDK sequentially binds to nine cyclins in three temporal waves [16,17]: Cln1-2 are expressed in G1 and mark the commitment to a new cycle [18–21], Clb5,6 promote DNA replication at S phase [22,23], and Clb1-4 drive progression through mitosis [24,25]. The fission yeast *Schizosaccharomyces pombe* has a single CDK that also binds different cyclins at G1,S, and M: Cig1,2 drive progression through G1 and S phase [26,27], and Cdc13 drives progression through mitosis [28,29]. However, a single CDK-Cyclin complex can drive progression through the entire cell cycle in this species [30].

In addition to the cyclin-CDK activity, the cell cycle is also regulated at the transcriptional level by timing the expression of genes required in its different phases. For instance, the E2F-Rb network of transcription factors controls initiation of DNA replication in animals at the G1/S transition [31–33]. In yeasts, transcriptional regulation of the G1/S transition is driven by SBF and MBF, two transcription factor complexes that bear no homology to E2F [34]. Recent findings show that these transcription factors were acquired through lateral gene transfer during fungal evolution [2]. In addition, other transcription factors are the mitotic Fkh1 and Fkh2 [35,36] in *S. cerevisiae* or the Hcm1 transcription factor, controlling progression through G2 and mitosis [37]. In human cells, a protein of the same family, FoxM1, also regulates gene expression in mitosis [32,38]. Although oscillatory transcriptional activity during the cell cycle is present in numerous species and cell types [32,33,37,39–50], the genes affected by cell cycle-regulated transcription are divergent between distantly related species

[51]. Likewise, even among different human cell types, periodic expression of only a fraction of genes is common to all of them [32,52].

Yeasts have historically been a powerful model system to understand the control of the cell cycle in animals. However, it has become clear that many otherwise conserved cell cycle regulators have been lost and independently evolved in the fungal lineage [2,3,53–55]. Thus, we sought to investigate the cell cycle control in another organism within opisthokonts that could potentially retain the cell cycle regulation program before it diversified in animals. We focused on *Capsaspora owczarzaki* (hereafter *Capsaspora*), a species more closely related to animals than yeasts, easy to culture, and for which good genomic resources are available [56–59]. This amoeba has a life cycle that includes three distinct stages that differ both in their morphology and transcriptional and proteomic profiles: amoebas with filopodia that proliferate in adherent cultures, an aggregative multicellular stage in which cells produce an extracellular matrix, and a cystic form that lacks filopodia [60–62]. Moreover, *Capsaspora* has a compact, well-annotated genome, with many homologs to animal genes [59]. With recent advances that allow transfection in the laboratory [63], *Capsaspora* is becoming a tractable model organism.

In this work, we have established a protocol to synchronize cell cycle progression in *Capsaspora* and have characterized its cell division and transcriptional profile across the entire cell cycle. We found that globally, the periodic transcriptional program of *Capsaspora* is enriched in genes that date back to eukaryotic origin, and it resembles human cells more than the periodic transcriptomes of yeasts. Out of four human cyclin types, *Capsaspora* contains homologs of cyclins A, B, and E. We found that these three cyclins are transcriptionally regulated during the cell cycle and have a conserved temporal order and cell cycle stage with human cells. In contrast, *Capsaspora* only contains one ancestral copy of the CDK, which likely form complexes with all of the *Capsaspora* cyclins. We also found that orthologs of many other cell cycle regulatory genes have a conserved timing of expression compared with animal cells. Our findings suggest that the cyclin-CDK system of animals evolved gradually, through an intermediate stage where one single CDK was able to interact with several cyclins at distinct stages of the cell cycle. Thus, while expansion and subfunctionalization of the cyclins found in animals occurred earlier throughout eukaryote evolution, we hypothesize that the expansion of metazoan cell cycle CDKs occurred concomitantly with the emergence of animals.

## Results

### Synchronization of cell cultures in *Capsaspora*

Synchronization of cell cultures is a basic experimental tool required to study the cell cycle [64]. Cell synchronization methods include temperature-sensitive strains, elutriation, cell sorting and commercially available inhibitors that arrest the cell cycle. Arrest and release approaches have previously been used to assess cell cycle progression in several organisms [44,47,65,66]. Hydroxyurea, a widely used S-phase inhibitor in yeast cells [67], was already shown to inhibit cell proliferation in *Capsaspora* cultures during the adherent cell stage [62]. Therefore, to check if cell cycle inhibition occurred before entering S phase and was reversible, we treated *Capsaspora* adherent cultures with hydroxyurea and assessed DNA content by flow cytometry. Upon hydroxyurea treatment, cells exhibited 1C DNA content, indicating arrest in G1 phase (Fig 1A). Upon wash and release into fresh media, we observed synchronous progression through the cell cycle as assessed by DNA content (Fig 1C). This indicates that hydroxyurea inhibits the cell cycle in S phase in *Capsaspora* and that its effect is reversible.

To measure the timing of cell cycle stages in *Capsaspora*, we treated two independently propagated cultures of *Capsaspora* adherent proliferative cells with hydroxyurea, and we later released them into fresh medium. Samples were taken at 45-minute intervals, starting from 2

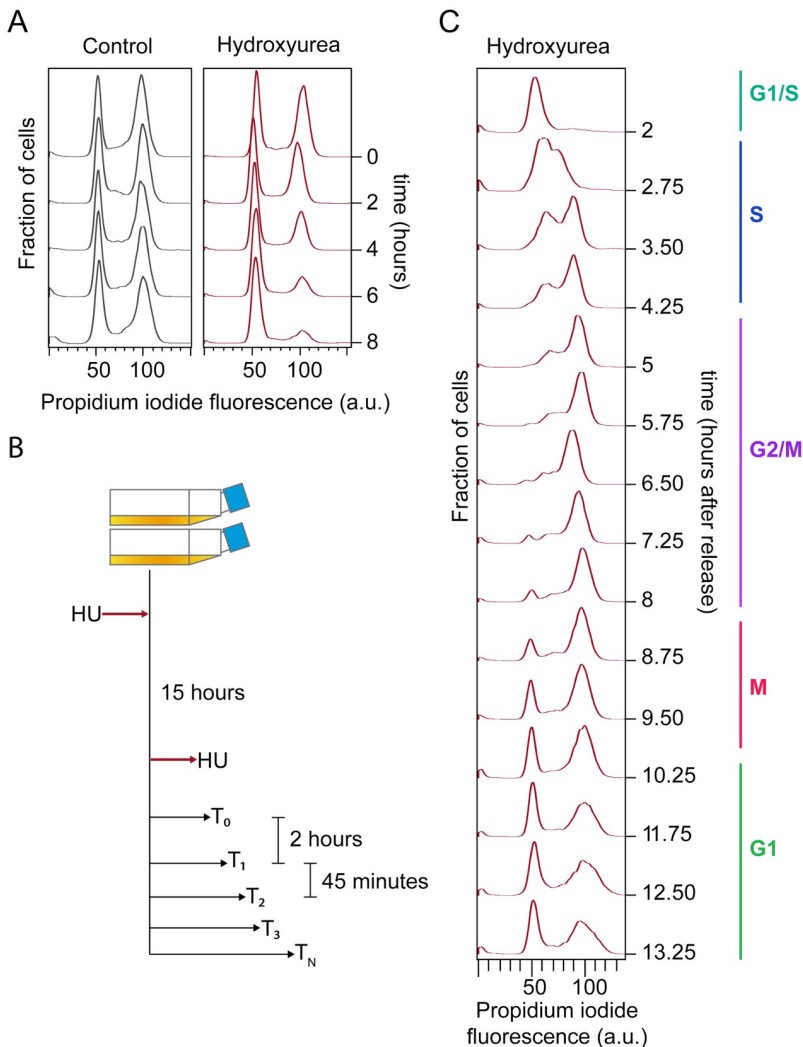

**Fig 1. Experimental setup for cell cycle synchronization in *Capsaspora*. A**: Effect of Hydroxyurea in *Capsaspora* cells. Control cells were treated with an equal volume of distilled water. HU treated cells were sampled every 2 hours, and a major decrease in 2C cells is observed after a minimum of 8 hours. **B**: Experimental layout used in this study. Two cell cultures growing independently for several generations were grown in fresh medium and kept in HU for a lapse of fifteen hours. After that, cells were released from HU and harvested every 45 minutes for 11 hours. Three different samples were taken from each culture at each time point. **C**: DNA content profile of synchronized *Capsaspora* cultures at representative time points of the whole experiment.

hours after release, taking a total of 16 time points that were analyzed using flow cytometry (Fig 1B). Following release, cells spent approximately 1.5 hours duplicating their DNA content, and after 8 hours from release a G1 peak appeared again, indicating completion of the cell cycle (Fig 1C). These observations were reproduced in the two replicates. At later time points, we noticed co-occurrence of 1C and 2C peaks. This may be due to some cells progressing through the cell cycle more rapidly than others [68], causing loss of synchrony, or due to an irreversible arrest in a fraction of the cells upon HU treatment [69]. Based on our observations of DNA content, *Capsaspora* synchronized cultures took ten hours approximately to double in cell density (Fig 1C), a time slightly slower than non-treated cultures of *Capsaspora* (S1 Fig). Nevertheless, our time course encompasses a complete round of the cell cycle, as we observed an increase in 1C at later time points.

## Dynamics and morphology of cell division in *Capsaspora*

To characterize the dynamics of cell division in *Capsaspora*, we used time-lapse microscopy in synchronized cells. In parallel, we analyzed synchronized cells for DNA content by flow cytometry and characterized the cell morphology during cell division using fluorescence microscopy. Out of a total of 100 mitotic cells observed by live imaging in the whole experiment (Fig 2A), the highest fraction underwent cytokinesis at approximately 10 hours (Fig 2C). Cells seemed to round up and slightly detach from the plate surface while retaining their filopodia (Fig 2A, S1 Video). This phenomenon has been previously characterized in other eukaryotic cells lacking a rigid cell wall, such as animal cells or *Dictyostelium* amoeboid cells [70–77]. Cells took an average of 3 minutes to completely undergo cytokinesis (Fig 2A, S1 Video), measured as the time from rounding up to the splitting of two daughter cells. As shown in Fig 2D, the measured area of daughter cells is roughly the same, suggesting that cell division is symmetric and yields two equally sized cells.

To investigate the morphology of mitotic cells, we stained tubulin and DNA. On each cell, we observed one dot of dense concentrations of tubulin, from which microtubules emanate (Fig 2B, white arrows). These dots duplicated and remained close, first associated with the nucleus and then surrounding densely packed DNA. A central, thicker spindle emerged and

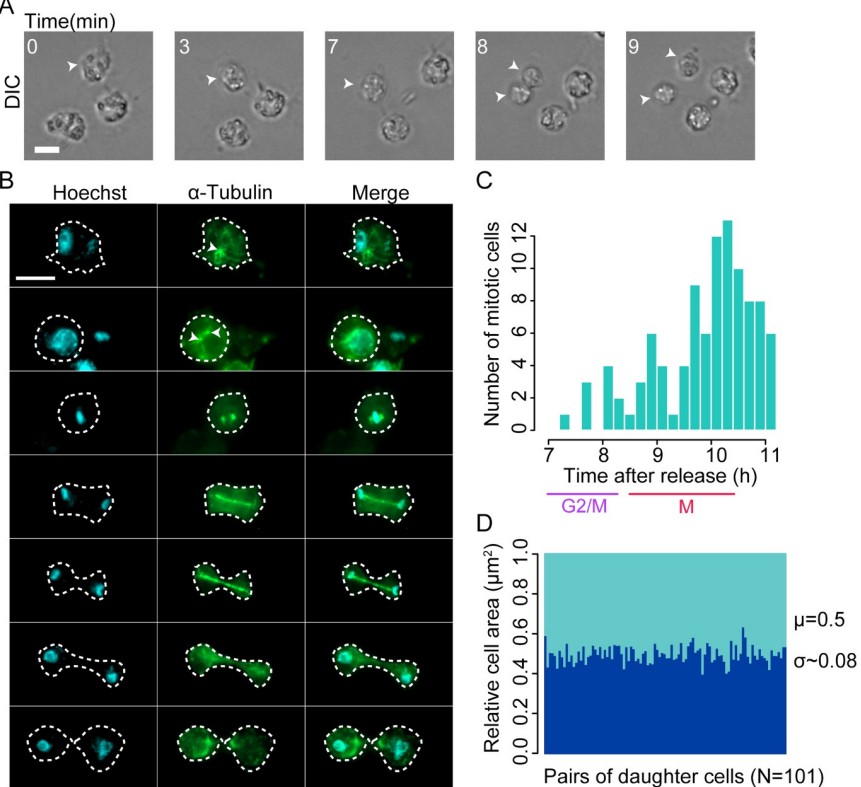

**Fig 2. Cell division in *Capsaspora*. A**: Time lapse of live imaging of a synchronous culture. Numbers indicate minutes since round up, with T0 still not rounded. White arrows indicate a cell dividing during the time lapse. Scale bar: 5μm. **B**: Fluorescence immunostaining of DNA (cyan) and Tubulin alpha (green) in *Capsaspora* synchronous cultures at different stages of cell division. White arrows indicate structures with a high concentration of tubulin. White dashed outline indicates cell perimeter. Scale bar: 5μm. **C**: Histogram depicting the number of cells at the moment of division in different times of the time lapse. Range goes from 7h to 11h. **D**: Stacked bar plot showing the normalized, relative cell area for each daughter cell in a total of 101 events of cell division.

grew as DNA separated and moved to opposite poles of the dividing cell. A similar phenomenon has been described for *Dictyostelium* [78]. These results came as a surprise, as a previous study has reported that *Capsaspora* lacks a centrosome and SPD-2/CEP192 homologs [79]. We further confirmed using BLAST best reciprocal hit that core human centrosomal proteins, as well as yeast spindle pole body proteins, are absent in several unicellular holozoan species, including *Capsaspora* (S2 Fig). These results suggests that *Capsaspora* mitotic spindle might be organized without a microtubule organizing center (MTOC) [80], or that it possesses an independently evolved MTOC, such as in yeast [81].

## Detection of periodically expressed genes during the *Capsaspora* cell cycle

In many cell types, cell division cycles are accompanied by a transcriptional program of periodic gene expression over time. To understand the transcriptome dynamics during the cell cycle of *Capsaspora*, we performed time-series RNA-seq experiments. We used RNA extracts from the same two biological replicates as in the flow cytometry assay and sequenced them using Illumina HiSeq v4. We processed the sequencing reads using Kallisto [82] (S3A Fig, S4 Text). Pearson correlations by gene expression profiles showed that time points are grouped according to the temporal order of sampling (S4 Fig). This indicates that gene expression is not shifted over time and was reproducible between the two replicates (Fig 3A). To detect periodicity patterns in gene expression, we applied two algorithms, JTK_CYCLE [83] and RAIN [84], on an average dataset of the two replicates where non-expressed genes were filtered out (S3B Fig). We assigned two ranks to every gene according to the p-values calculated by JTK_CYCLE and RAIN (Fig 3B), and assigned the final periodicity rank as the sum of JTK and RAIN ranks. We applied a conservative cutoff to identify genes that are undoubtedly periodically transcribed by taking the top 800 genes ranked within the top 2000 ranking for each independent dataset (Fig 3B) (S3B Fig). This cutoff corresponds to 10% of the total number of genes in *Capsaspora*, a fraction similar to those observed in other species [43,44,47,85]. Although false negatives with higher ranks might have been discarded due to our conservative approach, we confirmed that top-ranked genes showed oscillatory behavior by visual inspection. We manually included *Capsaspora* Cyclin A (CAOG_04719T0) [4,59] (see below) to the list of periodic genes despite ranking in position 1916, as it has a periodic expression profile (See below).

Expression of periodic genes correlates across replicates more strongly than non-periodic genes (Fig 3C) (S5A Fig). We also observed a strong correlation between initial and late time points (Fig 3C), suggesting that the cultures indeed completed the entire cell cycle despite the loss of synchrony. A principal coordinate analysis using data for the top-ranked periodic genes retrieved a grouping of time points that resembles a circle with two components explaining around 70% of the total distance (Fig 3D, S5B Fig), and a similar layout is obtained on a t-SNE projection of periodically expressed genes (S6C Fig). The results from these analyses indicate that there is a set of periodically transcribed genes during the cell cycle of synchronous *Capsaspora* cells. This set of genes represents the periodic transcriptional program of *Capsaspora* (S5 Text).

## Gene expression is clustered in periodic waves during the cell cycle

The cell-cycle-associated transcriptional program responds to the requirements of the cell at a given moment. For example, in many cell types, genes necessary for DNA replication or mitosis are transcribed only at the time of their biochemical activity. The detection of periodic genes in *Capsaspora* prompted us to classify them into temporal clusters, by centering their expression profiles to the mean and grouping them using hierarchical clustering based on a dissimilarity matrix by Euclidean distance (Fig 4A). Five clusters were detected according to the similarity in expression profiles over time (Fig 4A, S6A and S6B Fig), and we obtained very similar results

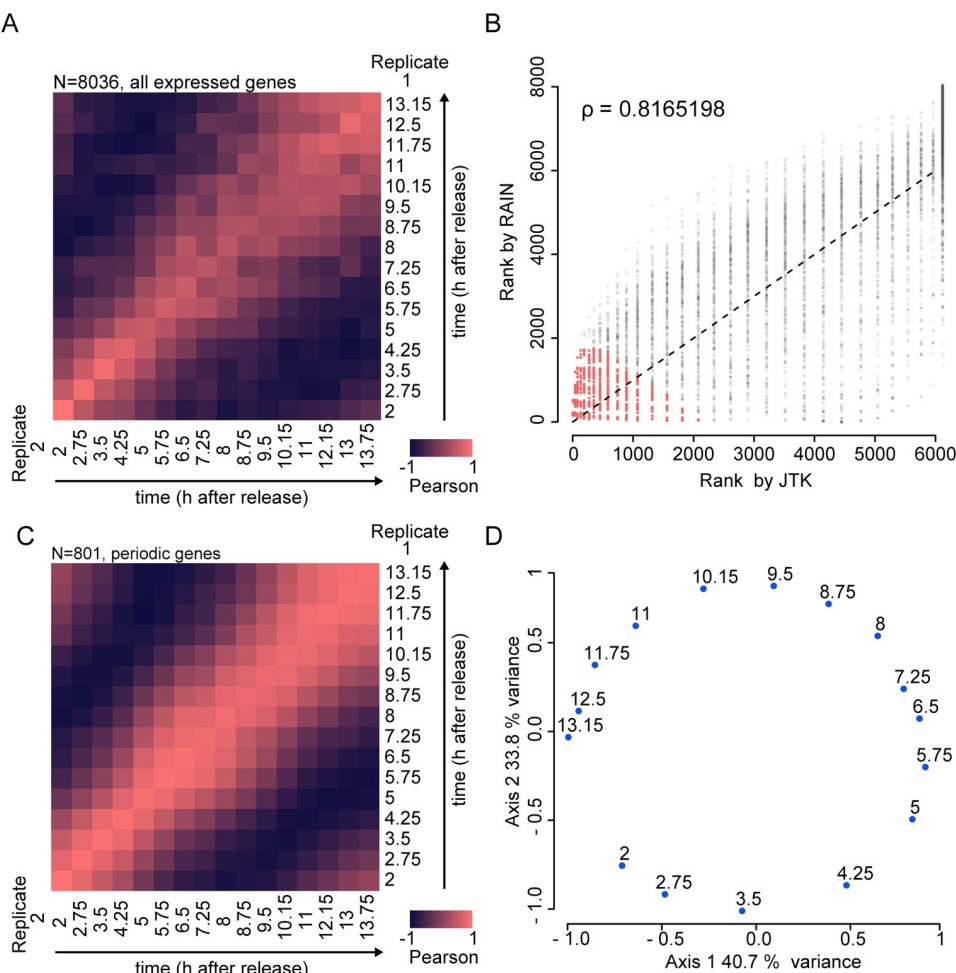

**Fig 3. Detection of periodic genes in *Capsaspora*. A**: Pearson correlation for the normalized expression level of every gene between replicates. Bright red indicates positive correlation, and dark purple indicates negative correlation. **B**: Spearman correlation between the two methods used to detect cell cycle regulated genes in *Capsaspora*. Scatter plots depicting the rank assigned for every gene by the JTK_CYCLE and RAIN on an average dataset of the two time-series replicates. These two algorithms rely on different approaches to finally assign, for every gene, a p-value interpreted as the probability that it can be considered periodic. For its proper functioning, we set the two algorithms to look for periodic behavior in a lapse of 11 to 13 hours, with time lapses of 0.75 hours. We assigned two ranks to every gene according to the p-values calculated by JTK_CYCLE and RAIN. Each dot represents a gene expressed in the time series. Colored dots represent the 801 genes that were finally taken as periodic according to our criteria. **C**: Pearson correlation for the top 801 cell cycle regulated genes, also based on normalized gene expression level. See Fig 4 for more details. **D**: Principal coordinate analysis on the set of 801 cell cycle regulated genes based on normalized gene expression level. Every dot represents a time point in the experiment.

by using k-means clustering (S7 Fig). We then associated these gene clusters to the cell cycle stage during the peak of expression, according to our DNA content and microscopy data: the G1/S cluster, S cluster, G2/M cluster, M cluster, and G1 cluster. Next, we calculated gene ontology (GO) enrichment for every cluster against the whole periodic transcriptional program using Ontologizer [86] Parental-Child-Union calculation (Fig 4B, S6 Text).

The G1/S cluster contains 194 genes peaking in the initial time points, from 2 to 3.5 hours after release. Genes found here exhibit the largest differences in gene expression between time points and are enriched in GO terms related to DNA replication, deoxyribonucleotide biosynthesis, and chromosome organization. There is also enrichment in the GO term "response to

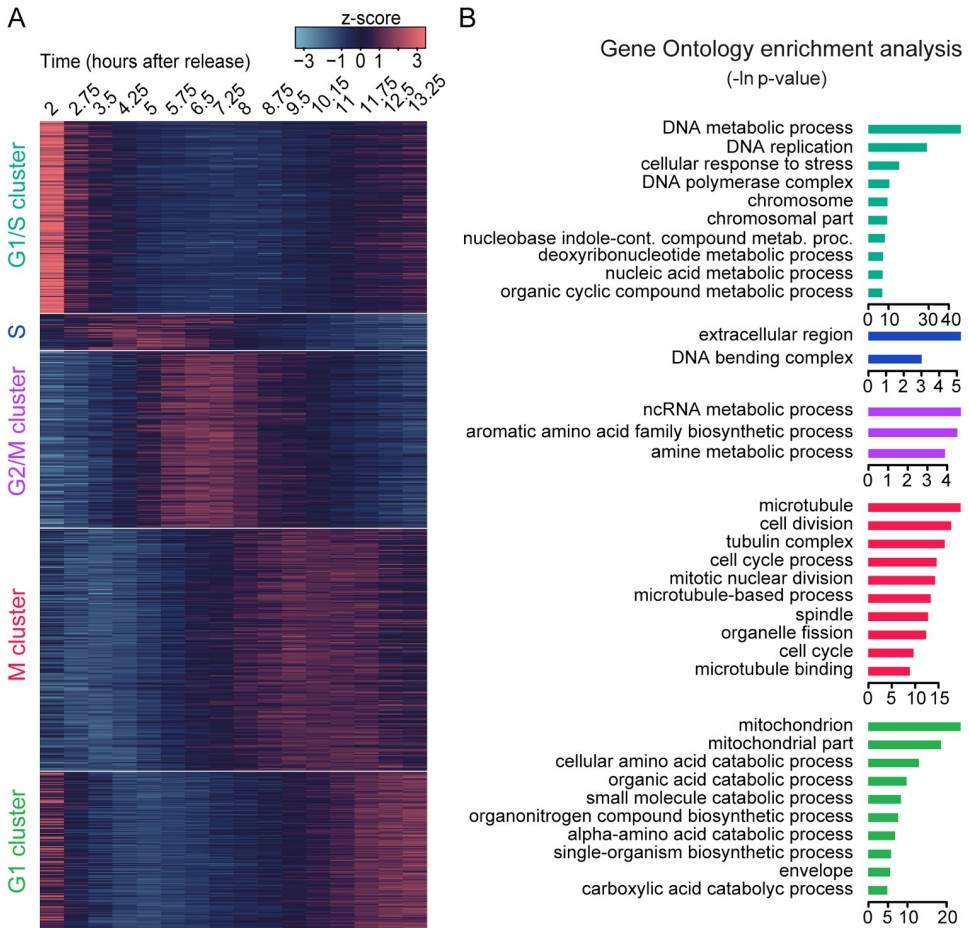

**Fig 4. The periodic transcriptional program of *Capsaspora*. A**: heatmap of gene expression level depicting the five main clusters detected by Euclidean distance hierarchical clustering. Clusters were rearranged to visually represent their expression peaks over time. **B**: Top ten enriched gene ontology terms for every cluster shown in A. We considered an enrichment as significant when Bonferroni corrected p-value was lower than 0.05. For the full list of enriched GO terms, see S6 Text.

stress", suggesting an effect of the treatment with hydroxyurea [68,69]. The small cluster assigned to S/early G2 contains 36 genes peaking between 3.5 to 5.75 hours enriched in the GO term "nucleosome binding" and includes several histone genes.

In the G2/M cluster, we found 176 genes with the peak of expression at 6.5–7.25 hours post-release. This cluster is enriched in the GO term "non-coding RNA metabolic process", and contains genes related to tRNA maturation such as RTCB, DDX1 [87], and tRNA ligases. It has been previously reported that tRNA synthesis can increase during the cell cycle in several systems [88–90]. To our knowledge, however, there is still no link reported between tRNA modification and progression through the cell cycle.

The M cluster is the largest one in the dataset, with 241 genes peaking between 9.5 and 11 hours. Genes reach the highest expression during time points when the cells enter mitosis, reaching a plateau which reflects the partial asynchrony of the cells at the time. This cluster is enriched for genes annotated with chromosome segregation, organelle fission, and diverse cytoskeletal components like spindle proteins and myosins. All these GO terms can be linked to mitotic cell division.

The G1 cluster has 157 genes. Genes in this cluster show higher expression levels in both the late and initial time points of the experiment. Many GO terms enriched in this cluster are related to the mitochondrion and diverse metabolic processes that indicate an increase in cell metabolism as the cell progresses through the cell cycle [91].

Taken together, the GO enrichment analyses show that gene expression clusters contain conserved genes involved in the cell cycle in *Capsaspora*.

## Conserved temporal order of cyclin and CDK expression in *Capsaspora*

In eukaryotes, the cell cycle events are regulated by cyclins in complex with CDKs. While the cyclin and CDK gene families are broadly conserved across eukaryotes [2,3], some of the sub-families are lineage specific and have radiated differently. In budding yeast, two types of cell cycle cyclins can be found: cyclin B and Cln-type cyclins. Both types bind to one single CDK, Cdk1 [16]. In animals, this ancestral CDK expanded and specialized [4] resulting in multiple cyclin-CDK partners involved in different phases of the cell cycle: CDK2 binds to cyclins E and A at the onset and later stages of S phase [13], cyclin B binds to CDK1 in mitosis [14], and CDK4 and CDK6 bind to cyclin D types during G1 [92]. To our knowledge, how cyclin-CDK binding partnership grew in complexity remains unclear.

To provide some insights into the evolution of cyclin-CDK binding partnership, we used our temporal gene expression dataset to infer the timing of cyclin-CDK activity during the cell cycle in *Capsaspora*. Due to inconsistencies in published work [4,59], we first revised the classification of cyclin and CDK genes found in *Capsaspora* by phylogenetic profiling using a complete taxon sampling for Holozoa (the clade comprising animals and their closest unicellular relatives) and validated the gene previously reported as *Capsaspora* CDK1 [59] by Sanger sequencing and transcriptomics (see Methods) (S8 and S10 Text, S8, S9 and S10 Figs). Despite the limited phylogenetic resolution, *Capsaspora* and other unicellular holozoan sequences appeared in earlier branching positions to the metazoan Cyclin A, B and E clades, this being compatible with *Capsaspora* having orthologs of these cyclins (S8 Fig). The phylogeny does not suggest the presence of a Cyclin D ortholog in *Capsaspora*. In the CDK phylogeny (S9 Fig), sequences from *Capsaspora* and other filastereans branch as a sister-group to the metazoan CDK1 clade (100% of UFBoot), whereas the branching of sequences from other unicellular holozoans is more uncertain concerning the CDK1 and CDK2-3 metazoan clades. From this, we envision two possible evolutionary scenarios. In a first scenario, an ancestral duplication of CDK1-3 into CDK1 and CDK2-3 occurred in a common ancestor of Holozoa. As most unicellular holozoans have only one sequence within the CDK1-3 clade, this would imply differential losses of either CDK1 or CDK2-3 in Ichthyosporea, Filasterea and Choanoflagellatea, and Metazoa conserving both paralogs. Despite *Salpingoeca rosetta* has two sequences within the CDK1-3 clade, both paralogs are likely to descend from a duplication event occurred in the Choanoflagellatea lineage, with *Monosiga brevicollis* losing one of the two copies. In a second scenario, the duplication would have occurred in the lineage leading to Metazoa, but the limiting phylogenetic signal would not have allowed reconstruction of the real phylogenetic pattern of the CDK1-3 clade. We find this scenario more parsimonious as it does not require any gene loss, whereas the other involves independent losses of an hypothetical paralog in all the holozoan lineages except Metazoa. This scenario is also supported by [2], where holozoan CDK1/2/3 is placed with plants CDKA. Thus, we propose that the subfunctionalization of CDK1-3 is a specific feature of Metazoa, with *Capsaspora* retaining the ancestral CDK1-3 gene instead of having a CDK1 ortholog as previously reported [59].

We report a clear temporal ordering of expression of the putative *Capsaspora* Cyclins A, B, and E (Fig 5A). Cyclin E belongs to the G1/S cluster, cyclin A clusters together with S-phase genes, and cyclin B is in the M cluster, peaking at mitosis (Fig 4) together with *Capsaspora* CDK1-3 (Fig 5B), although we found the CDK1/2/3 transcript expressed at high levels also during the rest of the cell cycle (S11B Fig). We also found periodic expression in G2 of a *Capsaspora* ortholog of the Cyclin G/I subfamily, with a known role as a G2/M repressor in human cells (Kimura et al., 2001), and a *Capsaspora* cyclin of uncertain phylogenetic relationship (CAOG_01199) (S11A Fig). We further confirmed the dynamic expression of *Capsaspora*'s CDK1/2/3, Cyclin B and Cyclin E using RT-qPCR (S11C Fig). In conclusion, our results suggest that cyclins A, B and E follow the same temporal order and cell cycle phases as cyclins in human cells.

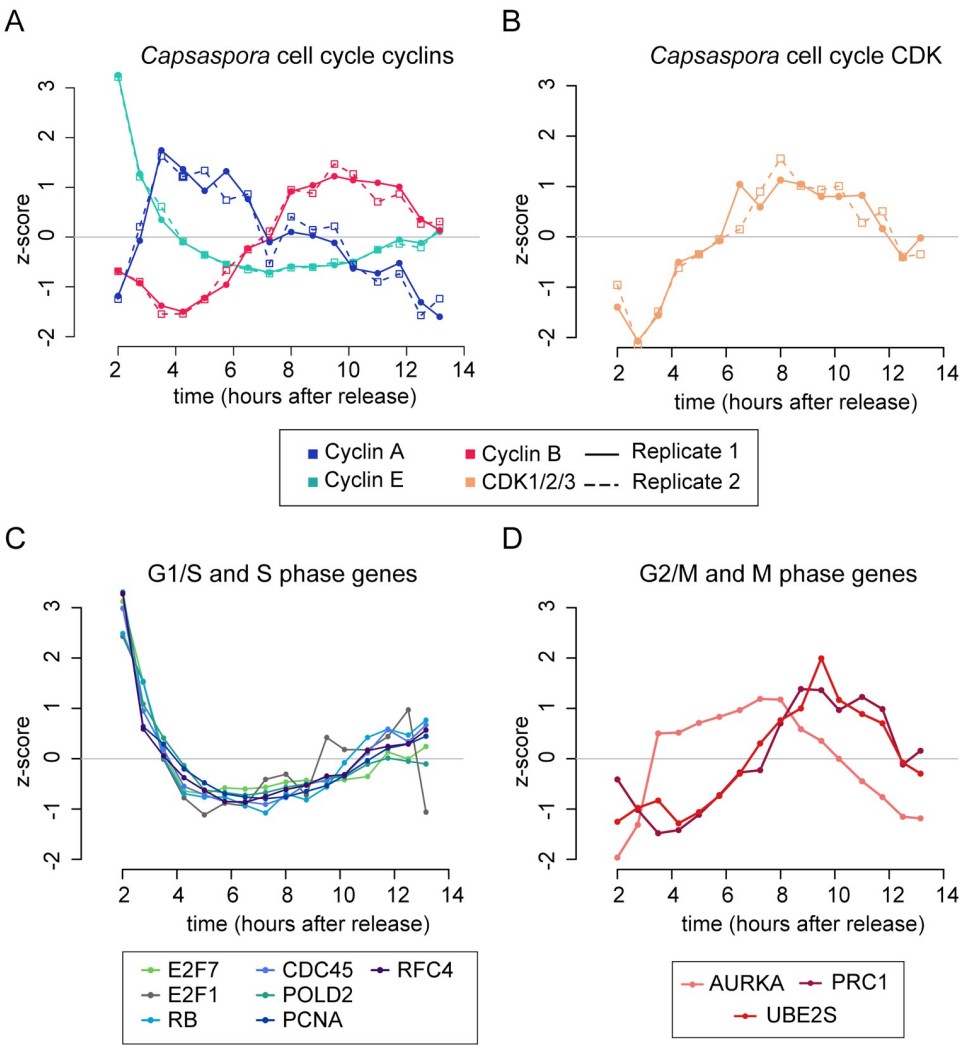

**Fig 5. Dynamics of the cyclin-CDK system and other cell cycle regulators in *Capsaspora*. A**: Gene expression profile of the Cyclin A, B, and E genes found in *Capsaspora*. **B**: Gene expression profile of the *Capsaspora* orthologue of CDK1/2/3 found by phylogenetic analyses (see S9 Fig). Normal and dashed lines indicate replicates 1 and 2, respectively. **C**: Gene expression level of *Capsaspora* orthologues of several G1/S regulators in animals. **D**: Gene expression level of *Capsaspora* orthologues of several G2/M regulators in animals. Genes in C and D as described by [32,44,45,93]. A full list is depicted in S12 and S13 Text.

## The *Capsaspora* periodic transcriptional program includes conserved eukaryotic genes and is similar to that of animal cells

Besides the cyclin-CDK system, other regulators are periodically expressed during the cell cycle [32,44,45]. To characterize the periodic expression program in *Capsaspora* in comparison to other species, we identified what are the *Capsaspora* orthologs of genes with known roles in regulating the cell cycle in other species. We did so by using OrthoFinder [94] (table 'Orthofinder' in S8 Text and using a list of one-to-one *Capsaspora*-human orthologs from a set of phylogenies of *Capsaspora* [61]. From these sources, we identified which periodic human genes with known functions in the cell cycle (as described in [32,44,45,93]) have also a periodic ortholog in *Capsaspora*. We found that numerous DNA replication genes are upregulated in the G1/S cluster in *Capsaspora*, including DNA polymerase subunits, replication factors, and proteins CDC45 and PCNA (Fig 5C). We also found periodic expression of the protein Retinoblastoma (Rb), one of the main regulators of entry into S phase in many eukaryotes [95]. *Capsaspora* Rb is temporally coexpressed with E2F7, a member of the E2F family of transcription factors known to act as a transcriptional repressor of the cell cycle [96,97] (Fig 5C). Among human genes that peak in mitosis, we found *Capsaspora* orthologs of Aurora Kinase A (AURKA), protein regulator of cytokinesis 1 (PRC1) and the anaphase-promoting complex (APC) subunit UBE2S expressed in the G2/M cluster (Fig 5D). In human cells, AURKA regulates the assembly of the centrosome and the mitotic spindle [98], mitotic cyclins and cohesins are degraded by the APC [1], and PRC1 regulates cytokinesis by cross-linking spindle midzone microtubules [99]. Although we did not find regulatory APC subunits CDC20 and CDH1 [100,101] among periodically expressed genes in *Capsaspora*, the UBE2S peak in mitosis suggests that APC activity might also be transcriptionally regulated during the cell cycle in *Capsaspora*. During M phase, we also observed upregulation of different kinesins, microtubule motors with conserved function in the mitotic spindle, and centromere proteins; a more detailed overview is provided in S12 Fig. We also identified *Capsaspora* periodic genes belonging to the same orthogroups as other cell cycle regulators in human cells, but without a one-to-one ortholog relationship (S13 Fig).

In addition to the examples above, we were interested in the similarities of the periodic transcriptional program of *Capsaspora* with those of other eukaryote species. First, to understand the evolutionary origin of the periodic genes found in *Capsaspora*, we calculated gene age enrichment for every cell cycle cluster. We assigned a gene age to the orthogroups by Dollo parsimony [102], resulting in ancient genes present in all species and younger genes found only in *Capsaspora* (see Methods). We compared the gene age enrichment ratios for non-periodic genes and the five clusters separately with the gene age of the whole transcriptome of *Capsaspora*. Three of the clusters of periodic genes presented significant enrichment in pan-eukaryotic genes (Fig 6A, S14 Text). Our data thus shows that a large fraction of genes in the periodic transcriptional program of *Capsaspora* belong to gene families originating early in eukaryotic evolution.

Next, we compared our dataset of *Capsaspora* periodic genes with datasets of cell cycle synchronized cells of different organisms, namely three different cell types of *Homo sapiens* (Hela Cells, U2OS cells, and foreskin primary fibroblasts) [32,43,46], *S. cerevisiae* [48], *S. pombe* [85], and *Arabidopsis thaliana* [49]. For each dataset, we took the published lists of periodic genes and corrected for the number of genes in each species (S15 Text). We set a threshold of less than 10% of genes to be periodic for human cells and the yeasts, and less than 5% for *A. thaliana*. Thus 1790 periodic genes were identified in HeLa cells, 1245 in U2OS cells, 461 in fibroblasts, 592 in *S. cerevisiae*, 499 in *S. pombe*, and 1060 in *A. thaliana* datasets. We found 1925 orthogroups that contained at least one periodic gene from either of the datasets (Fig 6B), and named these "periodic orthogroups". Of these, one third had orthologs in all five species.

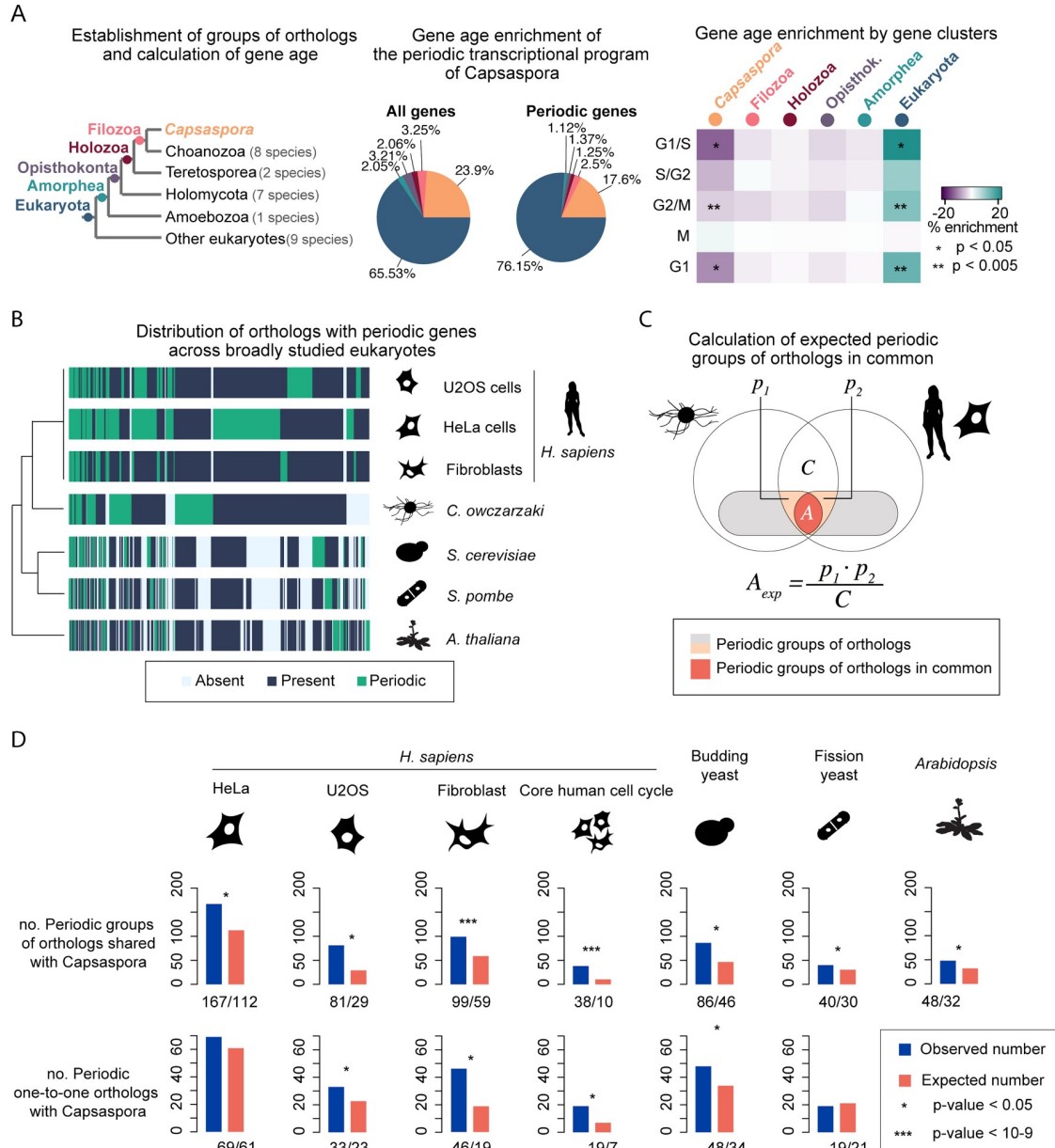

**Fig 6. The *Capsaspora* periodic transcriptional program has more resemblances to that of animal cells. A**: (Left) Phylogenetic tree of *Capsaspora* where colored nodes depict the different gene ages considered in this study. We used a database of 28 species of eukaryotes to infer groups of orthologs that were used to determine the gene age in *Capsaspora*. (Center) Gene age stratification profiles of the whole transcriptome of *Capsaspora*, as well as the periodic transcriptional program. (Right) Heatmap depicting gene age enrichment or depletion on each cluster of periodic genes in *Capsaspora* (see Fig 4), compared to non-periodic genes. Color scale indicates percentage of enrichment, and asterisks indicate the level of significance (Fisher's Test). **B**: Heatmap showing periodic orthogroups in common in the tree of species used in the comparative analysis. Every column represents a group of orthologs, which can be present, absent, or contain at least one periodic gene in each species. **C**: Venn diagrams indicating how the binomial tests were calculated for each pair-wise comparison of species, e.g. *Capsaspora* and *Homo sapiens* HeLa cells. White circles indicate orthogroups from each species. The intersection between these represents the orthogroups in common, *C*. *p1* and *p2* areas represent periodic genes of each species within *C*. Null expectation probability ($A_{exp}$) is calculated as the product of *p1* and *p2* divided by *C*. P-values of all the binomial tests are provided in S16 and S17 Text. **D**: Bar plots indicating the amount of shared periodic orthogroups and/or periodic one-to-one orthologues between pairs of cell types or species. P-values of all the binomial tests are provided in S17 Text.

We first computed the number of periodic genes shared between pairs of species (defined by their presence in the same orthogroup). Overall, all species share a small number of periodic genes (S16 and S17 Text), with the number of genes being highest for *Capsaspora* with *H. sapiens* cell lines (167, 81 and 99 periodic orthogroups with HeLa cells, U2OS cells, and fibroblasts, respectively) rather than yeasts or *A. thaliana* (Fig 6D, S15 and S16 Text). Still, the periodic expression of the majority of the genes is not shared between species, consistent with former findings of 2% to 5% of periodic genes shared between different organisms [51]. This indicates that the periodic transcriptional program is divergent both between species and even between different cell types within an organism.

Despite the low numbers of periodically expressed genes in common, we wondered whether the periodic transcriptional program in each species indeed evolved independently, meaning that the pairs of genes that share the periodic expression between species are observed only by chance. To that end, we calculated the expected number of shared periodic orthogroups by chance as the product of ratios of periodic orthogroups from each species within their orthogroups in common (Fig 6C). We detected that, for most species, the number of shared periodic genes is higher than by chance, especially for *Capsaspora* and the core cell cycle gene set of *H. sapiens* (defined in [32]) (Fig 6D, S16 and S17 Text). We found the same when comparing periodic one-to-one orthologs [61] (Fig 6D), when applying a cuttof of 20% to the periodic transcriptional program of *Capsaspora*, or when defining periodic genes on every species using the same method that we applied to *Capsaspora* (S13 and S14 Figs, S16 and S17 Text). Likewise, we found similar results when using an expanded dataset of 20% of the total transcriptome Therefore, these findings are robust with respect to the methods used to identify periodically expressed genes and to assign orthology relationships. Thus, although the cell cycle periodic expression program largely evolves fast and independently, our data suggest there is a core set of genes of conserved oscillatory expression during the cell cycle (Fig 6B). This set comprises 55 orthogroups that are periodic in the five species. These orthogroups contain genes such as cyclins and histones, and are enriched in common functions such as protein phosphorylation, DNA-binding protein, and cell division (S12 Fig) (S18 and S19 Text) (See Methods).

Overall, our cross-species comparison of the periodic gene expression programs revealed that the *Capsaspora* periodic gene expression program is more similar (in terms of number of periodic orthologs) to human cells that to current unicellular model systems for the cell cycle. Furthermore, including a new species in the global analysis, we discovered a previously unappreciated core set of genes for which periodic expression is deeply conserved.

## Discussion

In this study, we have used synchronized *Capsaspora* adherent cells to gain insight into key aspects of the cell cycle, such as cell division and periodic gene expression, in a unicellular relative of animals. Previously, the cell cycle has been studied in only a handful of species due to the inability to obtain synchronous cell cultures. With this synchronization protocol, the cell cycle of a closer relative of animals can be studied in cultures that can be synchronized from DNA replication to cell division.

Our experimental setup made possible to characterize mitotic cell division in *Capsaspora*, which we found relies on microtubule-based structures, as previously described in other eukaryote species [103]. Our observations suggest the presence of a putative non-centrosomal microtubule organizing center (MTOC) in *Capsaspora*, which raises new questions about the mechanisms of cell division in this species. As non-centrosomal MTOCs have independently evolved in many different species [104], it may well be that *Capsaspora* has a non-centrosomal,

independently evolved MTOC, or that their microtubules are able to self-arrange, as previously shown in other systems [80].

Synchronization of *Capsaspora* cell cultures allowed us to study transcription during the cell cycle using RNA sequencing. We identified five waves of gene expression across time, with most genes being expressed in the G1/S transition and in mitosis. As in previously studied organisms, these waves of transcription can be grouped in clusters containing genes related to the main events of the phases of the cell cycle. The periodic transcriptional program of *Capsaspora* is enriched in genes that emerged at the onset of eukaryotes, showing that the cell cycle relies in numerous genes, such as DNA replication proteins and cytoskeleton components, which are common to all eukaryotes due to their roles in fundamental cellular processes. Although transcriptional activity during the cell cycle is present in numerous species and cell types [51], the genes affected by cell-cycle-regulated transcription are divergent between distantly related species, likely due to the fact that transcriptional regulation adapts to the environment and lifestyle of each particular cell type. Our observations suggest that this occurs largely by old genes gaining and losing periodic regulation, rather than new species-specific genes evolving to be regulated during the cell cycle. This is consistent with Jensen et al.'s observations that periodicity in complex activity can evolve rapidly in different lineages by recruiting different partners of the same complex, which preserves the periodic regulation of the entire complex [51]. Still, in contrast to this previous analysis with a more limited set of species [51], our analysis clearly revealed a core set of genes of which the periodic regulation is deeply conserved among eukaryotes.

From the temporal order of gene expression of *Capsaspora* cyclins, we conclude that cyclins A, E, and B follow the same order and are associated with the same cell cycle stage as in *H. sapiens* cells. In contrast to human cells, where these cyclins bind their respective partner CDKs, *Capsaspora* only possesses one ancestral CDK1-3, which, although periodically expressed with a peak in M-phase, exhibits high transcript levels throughout the cell cycle. Due to this, we propose that CDK1-3 might be the binding partner of cyclin A, B, and E in *Capsaspora* and that it might be involved in all phases of the cell cycle (Fig 7). Nevertheless, in the absence of biochemical data, we cannot exclude the possibility that *Capsaspora* cyclins A and E bind other non-canonical CDKs. Interestingly, in knock-out mice where all CDKs except for CDK1 were deleted, all cyclins were also found to bind CDK1 [105], which suggests that animal cyclins A, B and E are also able to bind CDK1.

Based on phylogenies made in S8 and S9 Figs (S8 and S9 Figs), and phylogenies previously made by [2], the last eukaryotic common ancestor (LECA) likely already contained cyclins A and B, and a single cyclin-dependent kinase. In contrast, deep-branching eukaryotic lineages do not contain cyclins D and E (note that although named as cyclin D, Archaeplastida cyclin D is phylogenetically related to metazoan cyclin G/I in [2] and our analysis) (S8 and S15 Figs). In the fungal lineage, well-studied organisms such as *S. cerevisiae* secondarily lost some of the components of the system. In Holozoa (the lineage that contains animals and their closest unicellular relatives), the ancestral cyclin-CDK system underwent further expansion and specialization, which in humans contains cyclins A, B, D and E forming discrete complexes with multiple CDKs acting at distinct cell cycle stages. Although the molecular biology of the cell cycle has mostly been studied in human cell lines, phylogenetic data suggest that the human cyclin-CDK system is conserved in all animals [4]. Unicellular Holozoa (which include ichthyosporeans, filastereans and choanoflagellates) contain only the ancestral CDK, but already contain cyclin E in addition to the other two ancestral cyclins. However, cyclin D also appears to be an animal innovation. Importantly, our work in *Capsaspora* shows that the cyclins A, B and E are already present during the same cell cycle phases as human counterparts.

**Fig 7. A model of the dynamics of the Cyclin/CDK system in *Capsaspora*.** Bold lines indicate cyclin levels and dashed lines indicate CDK levels across the cell cycle. Independent phylogenetic studies suggest that a diversified set of cell cycle cyclins A and B existed prior to the last common ancestor of eukaryotes (LECA). Some of these components were secondarily lost and independently expanded in different eukaryotic lineages, such as in fungi (which includes *S. cerevisiae*). Based on our phylogenetic and transcriptomic analysis, a likely scenario is that several cell cycle cyclins may be binding to a single CDK that is not transcriptionally stable in *Capsaspora*. This suggests the expansions of cell cycle cyclins predate the expansion of cell cycle CDKs, which later acquired specific binding to cell cycle cyclins in animals.

Given these data, a likely evolutionary scenario is that cyclins underwent duplication and subfunctionalization before the emergence of animals, acquiring regulatory roles during distinct cell cycle phases while binding to the unique CDK. This evolutionary intermediate state is present in *Capsaspora*. During the emergence of animals, CDKs underwent expansion and subfunctionalized to bind specific cyclins, forming discrete, active complexes at distinct stages. This suggests that there was a gradual evolution of the cyclin-CDK control of the cell cycle in the holozoan lineage during the emergence of animals.

## Materials and methods

### Cell cultures and culture synchronization

*Capsaspora* cells were incubated at 23°C in ATCC medium 1034 (modified PYNFH medium). Two cultures of *Capsaspora* independently propagated (coming from a culture split two passagings before) at 30–50% confluency were treated using 10mM Hydroxyurea (Sigma Aldrich, Saint Louis, MO, USA, #H8627) in culture medium, and left incubating for approximately 14 hours. After treatment, cells were washed by: (i) centrifuging at 5000G for 5', (ii) discarding the supernatant, (iii) resuspending in 15ml of fresh medium, (iv) centrifuging at 5000G for 5', (v) discarding supernatant, (vi) eluting in fresh medium for a final cell density of ~1 million cells/ml. From this, cells were seeded in 5ml flask cultures for time-point sampling. Samples were collected by scraping and washing two hours after release, and from there on every 45

minutes until thirteen hours. A total of sixteen time points were taken, constituting a time window comprising one event of genome duplication and one mitotic division.

We also tested different concentrations and incubation times of hydroxyurea, nocodazole (Sigma-Aldrich, #M1404), and aphidicolin (Sigma-Aldrich, #A0781). Only 10mM hydroxyurea for longer than thirteen hours showed arrest of the cell cycle, while Nocodazole had no observable effect by DNA content measurement, and the rest ruined the samples due to insolubility of the compound.

### Doubling time experiment

Doubling time of *Capsaspora* was measured using optical density (OD600). Briefly, three independent cultures of *Capsaspora* were seeded at a cell density of ~1M cells/ml. Every six hours, cells were scraped and optical density (OD600) was measured using an Eppendorf BioPhotometer model #6131 (Eppendorf Corporate, Hamburg, Germany, EU) with *Capsaspora* medium as a blank. Note that, due to time constraints, time point 18h was not measured. DNA content was also measured in parallel (Methods, see below). For each replicate, OD600 data points were fitted to a logarithmic curve. The reported doubling time was estimated as the mean doubling time calculated from the different fitting curves from each replicate.

### DNA content measurement

Cells were washed in PBS and fixed in 70% ethanol in PBS, then incubated in RNAse A (Sigma-Aldrich, #R6148) (one volume in three volumes of 1xPBS) for 24 hours at 37 ˚C. Cells were then incubated in a final concentration of 20μg/ml propidium iodide (Sigma-Aldrich, #P4170-25MG) for 72 hours at 4 ˚C. Samples were analyzed by flow cytometry using a BD LSR Fortessa analyser (Becton Dickinson, Franklin Lakes, NJ, USA). SSC-A and FSC-A were used to detect populations of stained cells. Single cells were gated by FSC-H and FSC-A. An average of 10,000 events per sample were recorded. PI-positive cells were detected using a 561 nm laser with a 610/20 band pass filter (red fluorescence). To estimate the cell count, Texas Red-A was plotted as histograms using FlowJo 9.9.3 (FlowJo LLC, Ashland, OR, USA).

### Cell microscopy and image analysis

Microscopy pictures were taken using a Zeiss Axio Observer Z.1 Epifluorescence inverted microscope equipped with Colibri LED illumination system and Axiocam 503 mono camera (Carl Zeiss microscopy, Oberkochen, Germany). A Plan-Apochromat 100X/1.4 oil objective (Nikon Corporation, Tokyo, Japan) was used for imaging fixed cells. For the live imaging, we used an EC Plan-Neofluar 40x/0.75 air objective (Carl Zeiss microscopy).

Image analysis was done using ImageJ software [106]. For fixed cells, we used the oval selection tool to draw the contour of each cell and measured cell perimeter. As cells are spherical, we computed cell area using ImageJ. We estimated the relative cell area of every pair of daughter cells by dividing each measurement by the sum of the two daughter cells areas. All the calculation and data plotting was done in R Software ver. 3.4.4 [107].

### RNA isolation and sequencing

Time point samples were washed in 1xPBS, poured in Trizol, and frozen at -80˚C. Total RNA was purified using Zymo RNA miniprep kit (Zymo Research, Irvine, CA, USA, #R2050). mRNA libraries were prepared using the TruSeq Stranded mRNA Sample Prep kit (Illumina, San Diego, CA, USA, Cat. No. RS-122-2101). Paired-end 50bp read length sequencing was

carried out at the CRG genomics core unit on an Illumina HiSeq v4 sequencer, with all samples from the same replicate being pooled in the same lane.

*Capsaspora* adherent cultures cDNA was obtained by RT-PCR using SuperScript III Reverse Transcriptase (Invitrogen, Carlsbad, CA, USA, #18080044) following the manufacturer's instructions. PCR was performed using Phusion High-Fidelity DNA Polymerase (New England Biolabs, Ipswich, MA, USA, #M0530L) following the manufacturer's instructions.

### Transcriptomic analysis

RNA reads were mapped using Kallisto v0.43.1 [82] using default parameters onto a set of the largest isoforms of the *Capsaspora* transcriptome [62]. The resulting time-series transcriptome in transcript-per-million (tpm) units is available in S4 Text. We retrieved only the transcripts whose average expression level is above 1 tpm in the whole time series.

Gene expression level was normalized by subtracting the mean over time and dividing by the standard deviation. Normalized datasets were clustered according to their Spearman correlation values using hierarchical clustering (R gplots library [107,108]). All the supporting information and raw datasets can be found in FigShare: https://figshare.com/s/4d642c9854efe6d879a7.

### Identification of periodically expressed genes

Periodic genes were detected in *Capsaspora* by ranking using JTK_CYCLE [83] and RAIN [84] on the time-series transcriptomes and an average of the two replicates. We set JTK_CYCLE parameters to periods = 14:16 time points (please note that they do not correspond to hours) and sampling interval = 0.75, and ranked every gene by their BH.Q value. We set RAIN parameters to period = 16 and delta = 0.75, and ranked every gene by their Benjamini-Hochberg corrected p-value. We set a cutoff of the genes ranked below 2000 on each separate replicate and simultaneously ranked below 800 in the average dataset (see S3 Fig, S5 Text).

### Clustering analysis

Periodic genes were hierarchically clustered according to similarity of gene expression over time (averaged between two replicates), and clustered by k-means clustering [107] using standard parameters and k = 5. Agreement between clustering methods was calculated as the number of genes belonging to the same pair of clusters divided by the size of the smallest cluster in the pair. Cluster membership can be found at S5 Text.

### Calculation of Gene ontology enrichment

Gene ontology enrichment was calculated using Ontologizer [86] using the–c "Parent-Child-Union"–m "Bonferroni" options. Bonferroni-corrected p-values were taken as significant when below 0.05.

### Identification of microtubule organizing center proteins in unicellular Holozoa

We used a list of MTOC proteins from [109,110] to retrieve the respective *H. sapiens* and yeast sequences from UNIPROT. We used BLAST+ v2.3.0 [111,112] best reciprocal hit with parameter -evalue 10e-5 using this list as a query to identify putative homologous proteins in the proteomes of *Salpingoeca rosetta* (Sros), *Monosiga brevicolis* (Mbre), *Ministeria vibrans* (Mvib), *Capsaspora owczarzaki* (Cowc), *Creolimax fragrantissima* (Cfra), *Sphaeroforma arctica* (Sarc), *Corallochytrium limacisporum* (Clim), *S. cerevisiae* (Scer), *Neurospora crassa* (Ncra), and

*Dictyostelium discoideum* (Ddis). Hits were considered significant when the e-value score was below 10E-5 for both searches. Sequences and BLAST matches can be found in S1, S2 and S3 Text. All the supporting information can be found in FigShare: https://figshare.com/s/4d642c9854efe6d879a7.

## Phylogenetic classification of CDKs and Cyclins from early-Holozoa taxa

Annotated sequences of cyclins and CDKs from *H. sapiens* and *S. cerevisiae* were retrieved from Cao et al. [4] and Swissprot [113], and *A. thaliana* sequences were taken from [113–115]. These sequences were used as queries to detect potential CDKs and cyclins orthologues in our dataset (S7 Text) using BLAST+ v2.3.0 [111,112]. Those sequences that aligned were BLASTed against a database including all proteins from *H. sapiens*, *S. cerevisiae* and *A. thaliana*. We only included in our phylogeny those sequences whose best hit against this database matched the original sequences used in the detection step. Proteins were aligned with MAFFT v7.123b. [116], using the -einsi option, and alignments were trimmed using trimAl v1.4.rev15 [117] with the -gappyout option. Trimmed alignments were manually inspected and cleaned of poorly informative sequences except if that sequences corresponded to early-branching holozoa (ebH) taxa. Cleaned alignment were used as inputs for phylogenetic inference with IQ-TREE v1.6.7 [118], using the -bb 1000 and -mset LG options. For CDKs, since the tree topology did not show any ebH sequences belonging to CDK families without orthologues in Metazoa, we performed a second phylogenetic inference using only the ebH and metazoan proteins to reduce the potential phylogenetic noise that may be introduced with the addition of non-informative divergence.

Sequences were classified into CDK/Cyclins families taking into account the tree topology and the UFBoot nodal support values [119]. Families were named according to the orthology relationship between the ebH protein and the *H. sapiens* sequence. For example, Sarc_g11690T was classified as CDK10 because its position in the tree suggests an orthology relationship to this *H. sapiens* protein (S9 Fig), whereas Cowc_CAOG_08444T0 was classified as CDK11-CDK11B because it is orthologue to both *H. sapiens* proteins (S9 Fig). We found three ebH cyclin subfamilies without orthologues in Metazoa, which were named accordingly to the corresponding orthologue in *S. cerevisiae* (PCL1, PCL2, PCL9; PCL5, CLG1; and PCL6-PCL7). Those ebH sequences showing ambiguous or poorly supported branching patterns were classified as uncertain.

We found that the current *Capsaspora* sequence reported by [59] as a CDK1 orthologue (CAOG_07905) is considerably shorter in length than other CDK orthologues. This could explain why it was not detected in the work by Cao et al. [4], if an e-value constraint was taken into consideration. By inspection of the genomic sequence from [120], we detected an assembly gap of 1kb neighboring the 3' end of the predicted annotation of CAOG_07905 (S10A Fig). In silico translation of the surroundings of this gap revealed protein domains conserved in H. *sapiens* and S. *cerevisiae* CDK1 proteins (S10A–S10C Fig). Upon realization that the gene annotation of *Capsaspora* CDK1-3 was incomplete, we designed forward and reverse primers (S10 Text) to PCR-amplify the unknown sequence from both genomic DNA and cDNA (S10A and S10B Fig). Sanger sequencing of the amplified products revealed one more intron and one more exon, which were used to reconstruct the missing sequence in the assembly. We mapped the transcriptome of Replicate 2 –timepoint 9 against this reconstructed sequence using tophat2 [121] with standard parameters. Identification of paired-end overlapping reads in the reconstructed exons using Tablet [122] verified the results of Sanger sequencing (S10A Fig). The updated cDNA and protein sequence of *Capsaspora* CDK1-3, which aligns much better with human and yeast sequences (S10C Fig), can be found in S10 Text.

## Real-time quantitative PCR

Three independent cultures of *Capsaspora* were treated with or without hydroxyurea (10mM) over 13 hours. Upon HU release, all cultures were seeded at a cell density of ~1M cells/ml. Samples were taken every two hours for a total period of twelve hours, taking one ml for DNA content measurement (see above) and another ml for RNA extraction (see above). cDNA preparation was adapted from a protocol on *Sphaeroforma arctica* (http://dx.doi.org/10.17504/protocols.io.wqdfds6). Briefly, cells were lysed on 400 uL TRIzol (Ambion) and RNA was isolated using a protocol of 1-bromo-3-chloropropane for phase separation and Isopropanol for precipitation, followed by DNAseI treatment (Roche) and LiCl-ethanol precipitation to eliminate traces of genomic DNA. cDNA was produced using oligo dT primer and SuperScript III retrotranscriptase (Invitrogen) following the manufacturer's instructions. cDNA was quantified using AceQ SYBR qPCR Master Mix (Vazyme Biotech) in a iQ cycle and iQ5 Multi-color detection system (Bio-Rad). Primer sequences are available at S9 Text. The total reaction volume was 20 μL. All reactions were run in duplicate. The program used for amplification was: (i) 95˚C for 3 min; (ii) 95˚C for 10 s; (iii) 60˚C for 30 s; and (iv) repeat steps (ii) and (iii) for 40 cycles. Real-time data was collected through the iQ5 optical system software v. 2.1 (Bio-Rad). Gene expression levels are expressed as number of copies relative to the transcript CAOG_09639T0, used as housekeeping. For visualization, expression across time-points was normalized as a z-score. Ct data from each experiment can be found at S11 Text.

## Identification of orthologous genes

Groups of orthologs were generated using OrthoFinder v2.1.2 [94] using default parameters (evalue 10e-3, MCL clustering) on a dataset of proteomes found in Table 'Orthofinder' in S8 Text [113,123–125].

One-to-one orthologues were taken from a phylome of 6598 genes of *Capsaspora* reconstructed in [61] using the algorithm by [126] (http://phylomedb.org/phylome_100).

## Determining cell cycle-regulated genes in H. sapiens, S. cerevisiae, S. pombe and A. thaliana

We downloaded the lists of cell cycle regulated genes which can be found at [32,43,46,48,49,85]. We translated the gene and probe names of their datasets using BioMart [127] and Uniprot [113] tools (S15 Text).

## Gene age enrichment analysis

We used Count software [102] to assign the emergence of every orthogroup–and therefore every *Capsaspora* gene- to a given ancestor in common between species by Dollo Parsimony. We defined six different ages for *Capsaspora*: 0 "*Capsaspora*-specific", 1 "Filozoa", 2 "Holozoa", 3 "Opisthokonta", 4 "Unikonta", and 5 "Paneukaryotic". All *Capsaspora* genes unassigned to any orthogroup were defined as "*Capsaspora*-specific". Gene age enrichment was calculated using contingency tables and significance by Fisher exact test using R software ver. 3.4.4. [107], and was corrected for multi-test hypothesis using Bonferroni correction.

## Comparative analysis

Every periodic and non-periodic gene of each of the seven datasets were assigned to their respective orthogroup, if any. We obtained seven lists of orthogroups containing periodic genes of each species or cell type, and also defined lists of orthogroups containing genes (regardless of periodic) of each of the five species. A subset of periodic orthogroups (those

containing at least one periodic gene from at least one of the seven datasets) was generated, and plotted for periodicity, presence, or absence, in all the datasets using R gplots library [108].

Numbers and ratios of periodic shared orthogroups and one-to-one orthologues, as well as binomial test p-values (see Fig 6D, S14, S16, S17 and S18 Text), were calculated using R software ver. 3.4.4. [125]. For each pair-wise comparison of species, e.g. *Capsaspora* and *Homo sapiens* HeLa cells (Fig 6C), we took the number of orthogroups they have in common as a total population, *C*. Then we looked at the number of periodic orthogroups from each species that are within this total population, *p1* and *p2*, and calculated the null expectation ($A_{exp}$) as a product of the ratios of these two subpopulations within the population of orthogroups in common.

## Characterization of the core set of periodic genes

For each investigated species in Fig 6B, we retrieved all the genes belonging to orthogroups with periodic genes in all the five species (measured as present in all and periodic in at least one dataset from every species) (named set-1). We also retrieved, for each species, all genes belonging to orthogroups present in all the five species (named set-2). For each species, we performed separate Gene Ontology enrichment analyses of set-1 against set-2, using Ontologizer with Bonferroni correction (see above). The significantly, most enriched p-values in all the species were retrieved and visualized using R (The R core team, 2018) (S18 Text). In parallel, we used an EggNOG v4.5.1 [128] annotation of the *Capsaspora* genome to retrieve the putative identity of the *Capsaspora* genes belonging to the orthogroups periodic in all the five species (S19 Text).

## Reanalysis of cell cycle datasets using JTK_CYCLE and RAIN

We reanalyzed the datasets of gene expression of [49] (HU treatment dataset), [85] (three replicates of elutriation), [48] (two replicates of wildtype synchronous yeast cultures), [46] (one replicate, thymidine block), [32] (four replicates of double thymidine block), and [43] (a dataset of ~8000 genes matching filtering criteria by the authors) using the same pipeline used in our *Capsaspora* datasets. For those with replicates, periodicity ranks were calculated for each replicate independently and summed at the end.

As every experiment comprised different numbers of cell cycles of different length, we set up JTK and RAIN parameters to look for periodicity in time lapses according to the author's reports (S15 Text). We corrected for the number of genes by setting a threshold of less than 10% of genes to be periodic. Overlap between datasets of periodic genes was calculated using R Software ver. 3.4.4. [107].

## Supporting information

**S1 Fig. *Capsaspora owczarzaki* doubling time. A**: Cell density curve (optical density, OD = 600 nm) at different cell concentrations of *Capsaspora* (three replicates per condition). Values of cell concentration around OD600 = 0.1 were used to seed cultures and calculate the doubling time of *Capsaspora* in B. **B**: Growth curve (logarithmic scale) of non-synchronized *Capsaspora* cell cultures (three replicates). Doubling time is calculated from three independent replicates.
(TIF)

**S2 Fig. Conservation of microtubule organizing proteins in unicellular Holozoa.** Presence/absence matrix of the animal and yeast microtubule organizing proteins across *Salpingoeca rosetta* (Sros), *Monosiga brevicolis* (Mbre), *Ministeria vibrans* (Mvib), *Capsaspora owczarzaki*

(Cowc), *Creolimax fragrantissima* (Cfra), *Sphaeroforma arctica* (Sarc), *Corallochytrium lima-cisporum* (Clim), *Saccharomyces cerevisiae* (Scer), *Neurospora crassa* (Ncra), and *Dictyostelium discoideum* (Ddis). List of proteins from Carvalho-Santos et al., 2011, Hodges et al., 2010, and Uniprot. Classification adapted from Carvalho-Santos et al., 2011, and Hodges et al., 2010.
(TIF)

**S3 Fig. Computational pipeline used to detect periodic genes in *Capsaspora*. A**: Pipeline for ranking the transcripts on each experiment. RNA reads were processed using Kallisto, noise transcripts were filtered out, and JTK and RAIN were run in parallel with the indicated setup. Bonferroni-corrected p-values were ranked, and the sum of ranks was used as a final rank. **B**: Pipeline used to detect periodic transcripts in *Capsaspora*. Samples were treated separately to detect periodic transcripts with a loose cutoff. This gene set was used to filter periodic genes ranked from an average dataset of the two replicates, out of which the top 800 (10% of the number of genes in *Capsaspora*) were selected.
(TIF)

**S4 Fig. Clustering of time points of each replicate and the unsynchronized culture sampled as a control, based on a dissimilarity matrix of Pearson correlation.** All the genes with an average expression level above 1 tpm in both replicates were used.
(TIF)

**S5 Fig. Reproducibility and informative value of *Capsaspora* periodic genes. A**: Distributions of Pearson correlation between replicates of a set of randomly chosen 801 genes and the 801 genes defined as periodic. **B**: Fraction of variance explained by the different relative eigenvalues of the principal coordinate analysis (see Fig 3D).
(TIF)

**S6 Fig. Details on the hierarchical clustering of *Capsaspora* periodic genes. A**: Hierarchical clustering of the expression profiles of the 801 periodic genes found in *Capsaspora*, and the color equivalences with the final clusters shown at Fig 4A. **B**: Average expression level of *Capsaspora* periodic genes grouped by hierarchical clustering. **C**: t-SNE plot of all 801 genes in the periodic transcriptional program of *Capsaspora*, showing circle pattern as in Fig 3D. Color code in both figures follows the color code in Fig 4.
(TIF)

**S7 Fig. K-means clustering of the periodic transcriptional program of *Capsaspora*. A**: Average expression level of *Capsaspora* periodic genes grouped by k-means clustering. **B**: Top ten enriched GO terms of each cluster of periodic genes generated by k-means clustering. GO terms were considered significant when Bonferroni-corrected p-value was lower than 0.05. Full list available at S3 Fig. **C**: Agreement between clustering methods. Heatmap showing the percentage of overlap between clusters by two methods. Overlap is calculated as the number of genes belonging to the same pair of clusters divided by the size of the smallest cluster in the pair.
(TIF)

**S8 Fig. Unrooted maximum likelihood phylogenetic tree (IQ-TREE) inferred from cyclins sequences of 30 eukaryotic species (see Methods).** Nodal support values (1000- bootstrap replicates by UFBoot) are shown in all nodes. Eukaryotic sequence names are abbreviated with the four-letter code (see Methods) and colored according to their major taxonomic group (see panel).
(PDF)

**S9 Fig. Maximum likelihood phylogenetic tree (IQ-TREE) inferred from CDK sequences of early-branching holozoan species and animals (see Methods).** Statistical support values (1000-replicates UFBoot) are shown in all nodes. Eukaryotic sequence names are abbreviated with the four-letter code (see Methods) and colored according to their major taxonomic group (see panel).
(PDF)

**S10 Fig. Re-annotation of the coding sequence of *Capsaspora* CDK1/2/3. A**: Schematic representation of the genomic locus of *Capsaspora* CDK1-3 gene showing exons, splicing sites, non-annotated regions of predicted sequence, and mapping of mRNA reads. **B**: PCR amplifications of *Capsaspora* CDK1 using primers detailed in Methods and A, using genomic DNA and cDNA as templates. Arrows indicate size of the products sent for sequencing. **C**: Alignment of H. *sapiens* and S. *cerevisiae* CDK1 genes, and the *Capsaspora* updated CDK1-3 sequence, using Geneious v8.1.9.
(TIF)

**S11 Fig. Expression of cyclins and CDKs in *Capsaspora*. A**: Gene expression profile of cyclin G/I and an unidentified cyclin gene (CAOG_01199) found in *Capsaspora*. **B**: Total amount of *Capsaspora* CDK1/2/3 throughout the cell cycle. **C**: Dynamics of the cyclin-CDK system using real-time PCR. Normalized gene expression profiles of cyclins B, E, and CDK1/2/3 in two independent biological replicates of synchronized *Capsaspora* cultures.
(TIF)

**S12 Fig. List of all enriched GO terms of the core cell cycle regulated gene set from each species used in Fig 6B.**
(PDF)

**S13 Fig. Analysis of the periodic transcriptional program of *Capsaspora* using a cutoff of 20% of the total transcriptome. A**: Scatter plot replicating Fig 3B, showing in blue all the genes additionally taken into account. Colored dots represent the 1600 genes that were finally taken as periodic in this reanalysis. **B**: Principal coordinate analysis using the dataset of 1600 genes. **C**: heatmap of gene expression level depicting six clusters detected by Euclidean distance hierarchical clustering. Clusters were rearranged to visually represent their expression peaks over time. Black arrow and dividing cell indicate time of cell division (see Fig 2). **D**: Gene ontology enrichment analysis of the six clusters represented in C. **E**: Bar plots indicating the amount of shared periodic orthogroups and/or periodic one-to-one orthologues between pairs of cell types or species, using the dataset of 1600 periodic genes in *Capsaspora*.
(TIF)

**S14 Fig. Reanalysis of previous cell cycle datasets in model organisms. A**: Scatter plots of ranks by JTK and RAIN for each dataset of each species used in the comparative analysis (see Fig 6D and Results section). Datasets were processed as indicated in S15 Text, and Material and methods. Depending on the dataset, we could recover between one third and more than half of the originally described periodic genes, except *Arabidopsis* where the agreement was very low. **B**: Bar plots indicating the amount of shared periodic orthogroups and/or periodic one-to-one orthologues between pairs of cell types or species, using our own lists of periodic genes. P-values of all the binomial tests are provided in S17 Text.
(TIF)

**S15 Fig. A possible reconstruction of the evolution of the cyclin-CDK system regulating the cell cycle in animals.** Phylogenetic tree of eukaryotes displaying the different expansions, gains and losses of the main cyclin and CDK subfamilies with a known role in cell cycle

regulation, in different eukaryotic groups. Events within groups refers to some species within the group showing additional expansions, gains or losses beside those depicted in the figure. Note that LECA cyclin G/I likely gave rise to plant cyclin D and share a common ancestor to the metazoan cyclins D and G/I. Gain of other cyclin families are depicted as regular letters at different stems.
(TIF)

**S1 Text. FASTA formatted sequences of human and yeast MTOC proteins used as query in S2 Fig.**
(FASTA)

**S2 Text. FASTA formatted sequences of the putative MTOC proteins from the organisms shown in S2 Fig.**
(FASTA)

**S3 Text. Tables of BLAST matches of human and yeast MTOC proteins in the proteomes of the organisms shown in S2 Fig.**
(TXT)

**S4 Text. Tables of transcript per million of each replicate.**
(TSV)

**S5 Text. List of periodic genes in *Capsaspora*, containing information for each gene about cluster membership, gene age, and orthologs in other species.**
(TSV)

**S6 Text. List of significant (Bonferroni p-value $< 0.05$) GeneOntology enrichments of each hierarchical cluster of periodic genes in *Capsaspora*.**
(XLSX)

**S7 Text. FASTA formatted sequences of cyclins and CDKs used in the phylogenetic analyses (see S8 and S9 Figs).**
(FASTA)

**S8 Text. List of species used in the phylogenetic analyses and in the generation of groups of orthologues.**
(XLSX)

**S9 Text. FASTA formatted sequences of the primers used in the sequencing of CDK1 and in the qPCR experiments.**
(FASTA)

**S10 Text. FASTA formatted sequences of newly annotated *Capsaspora* CDK1-2-3 CDS and protein translation.**
(FASTA)

**S11 Text. Data from the qPCR experiments.**
(XLSX)

**S12 Text. List of cell cycle regulators in humans (described in [32,44,45,93]), and their respective orthologs in *Capsaspora* defined by OrthoFinder and/or phylome data (see Results and Methods). Bold indicates genes that have been plotted in Fig 5C or 5D.**
(XLSX)

**S13 Text. List of cell cycle regulators in humans (described in [32,44,45,93]), that also have at least one periodic co-ortholog in *Capsaspora*, defined by OrthoFinder (see Results and**

Methods).
(XLSX)

**S14 Text. Metrics of gene age enrichment and depletion for the gene clusters of the periodic transcriptional program of *Capsaspora*, and their corresponding Fisher Test p-values.**
(XLSX)

**S15 Text. Procedure used to retrieve identifiers for the datasets of H. sapiens, S. cerevisiae, S. pombe, and A. thaliana, and parameters used to set up JTK_CYCLE and RAIN in the reanalysis.**
(XLSX)

**S16 Text. Metrics of shared periodic orthogroups (OG) and one-to-one orthologues between *Capsaspora* and the rest of cell types and species.**
(XLSX)

**S17 Text. Metrics of shared periodic orthogroups (OG) for all comparisons between pairs of species, using periodic genes from the literature and using our own sets of periodic genes.**
(XLSX)

**S18 Text. GO enrichment of human, yeasts, plants and Capsaspora gene sets belonging to the core conserved set of orthogroups in Fig 6B.**
(TXT)

**S19 Text. List of *Capsaspora* genes belonging to the core set of periodic orthogroups in Fig 6B, annotated using eggNOG (See Methods).**
(TXT)

**S1 Video. Synchronized cells of *Capsaspora* undergoing cell division.** Time interval between frames is 1 minute. The movie is played at 7fps. Scale bar = 5μm. Available on Figshare: https://figshare.com/s/4d642c9854efe6d879a7.
(AVI)

# Acknowledgments

We thank all the members of Iñaki Ruiz-Trillo's lab for discussion and comments on the manuscript, Sebastián Najle and Núria Ros for helpful advice and guidance with the Propidium Iodide staining and the qPCR experiments, Daniel Richter for discussion on the statistical tests, Xavier Grau-Bové for helpful discussions on the data analysis and for providing guidance in the Principal Coordinate analysis, Arnau Sebé-Pedrós for providing resources of the *Capsaspora* phylome, and Meritxell Antó-Subirats for technical and logistics support. We also acknowledge the UPF Flow Cytometry Core Facility for assistance with flow cytometry, and the CRG Genomics Unit for mRNA library preparation and Illumina sequencing.

# Author Contributions

**Conceptualization:** Andrej Ondracka.

**Formal analysis:** Alberto Pérez-Posada, Omaya Dudin, Eduard Ocaña-Pallarès, Andrej Ondracka.

**Funding acquisition:** Iñaki Ruiz-Trillo.

**Investigation:** Alberto Pérez-Posada, Omaya Dudin, Eduard Ocaña-Pallarès, Andrej Ondracka.

**Supervision:** Iñaki Ruiz-Trillo, Andrej Ondracka.

**Writing – original draft:** Alberto Pérez-Posada.

**Writing – review & editing:** Omaya Dudin, Eduard Ocaña-Pallarès, Iñaki Ruiz-Trillo, Andrej Ondracka.

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
