## [Decision Letter · Decision Letter 0]

21 Aug 2019

Dear Iñaki,

Thank you very much for submitting your Research Article entitled 'Gradual evolution of cell cycle regulation during the transition to animal multicellularity' to PLOS Genetics. Your manuscript was fully evaluated at the editorial level and by independent peer reviewers all of whom are experts in the area. The reviewers appreciated the attention to an important problem, but raised a series of points about the submitted manuscript. Based on the reviews, we will not be able to accept this version of the manuscript, but we will be happy to review again a revised version responsive to the reviews. On balance, this is between a minor revision and a major revision decision.

If you decide to revise the manuscript for further consideration at PLOS Genetics, please aim to resubmit within the next 60 days, unless it will take extra time to address the concerns of the reviewers, in which case we would appreciate an expected resubmission date by email to plosgenetics@plos.org.

[LINK]

We look forward to receiving your revised manuscript. Please do not hesitate to contact us if you have any concerns or questions.

Yours sincerely,

Joe

Joseph Heitman, MD, PhD

Associate Editor

PLOS Genetics

Bret Payseur

Section Editor: Evolution

PLOS Genetics

Reviewer's Responses to Questions

**Comments to the Authors**:

Reviewer #1: Synopsis: There is a growing interest in the cell biology of early-diverging holozoans (Monosiga, Salpingoeca, Sphaeroforma, and Capsaspora) because they provide insights on the early ancestor of animals. Here, the authors used hydroxyurea to block the cell cycle of Capsaspora and then synchronously release the population upon wash out (Fig. 1). They used microscopy to analyze cell division, tubulin staining, DNA straining of synchronized cells (Fig. 2) to quantify the synchrony and timing of the cell cycle (mitosis is 10.5 hours after release, total cell cycle period is ~12 hours). The authors then used RNA-seq to identify ~800 cell cycle regulated genes (10% of the genome) after synchronous block-release; Figs. 3-4. This dataset forms the basis of their bioinformatic analysis. The main claims are that Capsaspora cyclins E, A, B have similar timing to their animal counterparts (Fig. 5) and Capsaspora cell cycle transcription program resembles humans more than yeasts and plant (Fig. 6). Knowing the timing of cyclin and CDK gene expression in early-diverging holozoa is important for understanding the principles of cell cycle regulation and its conservation across eukaryotes.

Strengths: This work is novel because it is (to my knowledge) the first cell cycle expression dataset in an early-diverging holozoan. If so, the authors should state this more explicitly and put their work in context. I think the manuscript will be of broad interest to the cell cycle and evolutionary genomics community. The authors worked with a non-model organism and made great use of genomics, which should inspire other groups working with early-diverging holozoa or non-model organisms.

Weaknesses: The analysis of RNA-seq replicates is well done. The density of time sampling was high (45 minutes for 12 hours). The use of two methods for identifying periodic genes with 11 to 13-hour period (already known from Fig. 2) is good, and the circular structure in the two principle components (Fig. 3D) is reassuring. However, I could not find experiments that validate the timing of Cyclin E, A, B and Cdk1 expression observed in RNA-seq experiments, e.g. Fig. 5. The authors also make some claims regarding the expansion of cyclins and CDKs that are inaccurate and, thus, should be rewritten; see Comments below. Both of these weaknesses are relatively straightforward to address.

Major comments:

** Lines 197-201 and Discussion: Although it is exciting to see a mitotic spindle in the absence of a canonical MTOC, the absence of gamma-tubulin ring complex genes (i.e. evidence for lack of MTOC) could also be due to poor annotation or an incomplete genome? How confident are the authors that Capsaspora genome does not encode for canonical MTOC? Are Sphaeroforma, Monosiga, and Salpingoeca genomes also missing these components? If so, this would suggest that these genes are missing. Please elaborate and rewrite.

** One of the main conclusions is the conservation of gene expression timing of Cyclin E, A, B and CDK expression between Capsaspora and humans. However, I could not find experiments to validate the timing of Cyclin E, A, B and CDK expression observed in RNA-seq experiments. At a minimum, some alternative method to validate transcript levels (RT-qPCR, Northern blot) is needed. More ambitiously, one could measure Cyclin E, A, B protein levels using Western blot. This would really emphasize similarities between Capsaspora and humans, and improve significance and impact.

** According to the phylogenetic tree, Capsaspora lacks a Cyclin D, but it has a Cyclin F and two Cyclins G (which cluster with Cyclin E). Have the authors analyzed the expression profile of these cyclins in Capsaspora compared to humans? This would improve significance and impact.

** Ref. 2 has a Cyclin and CDK phylogeny (Figure 2-Figsupp2, Figsupp6) with a broad number of eukaryotic outgroups (including Capsaspora, Salpingoeca, Sphaeroforma, Monosiga). This reference will be helpful in two ways:

(i) Lines 318-332: The CDK phylogeny of Ref. 2 places holozoa CDK1/2/3 with "CDKA" (the ancestral CDK1/2/3 found in plants and deep protists). This provides better evidence in support of scenario 2 (CDK1 and CDK2/3 are metazoan-specific gene duplications).

(ii) Lines 482-486: LECA already had one CDK and several Cyclins (A, B, D/E/F/G/etc..). The evolutionary intermediate state (of multiple cyclins) is found in many other eukaryotes, not just Capsaspora; see Ref. 2. The sub-functionalization of cyclins happened between FECA (first common ancestor) and LECA. The authors should rewrite their discussion accordingly.

Minor comments:

** Line 187: I find this comparison awkward and meaningless (especially when comparing to yeast).

** Why is the JTK Rank clustered into vertical bands in Figure 3B? Please clarify.

** Line 254, Fig. 4: Are cells that are blocked in DNA replication due to hydroxyurea and then released still in G1/S? I ask because the GO enrichment categories are more consistent with S phase. Perhaps you could clarify in the main text how you arrived at these classifications. Is it based on DNA content and cell cycle events in Figs. 1-2 and/or GO categories? Perhaps label the inferred phases in Figs. 1-2?

** Line 297: "Cdc2" (old name, based on homology to Pombe) should be "CDK1"? The authors use CDK1,2,3,4,etc.. throughout their manuscript.

** Line 375: Could the authors add a sentence to describe how they calculate "gene age", i.e. gene found only in Capsaspora is youngest, gene found in all species is oldest?

** Line 419: Can you elaborate on the core set of genes of conserved oscillatory expression? This is novel and interesting.

** Line 1144: "Seven" should be "five".

Reviewer #2: The manuscript by Perez-Posada and colleagues describes the transcriptional regulation of the cell cycle in the unicellular protist Capsaspora and compares its cell cycle-related gene content (especially the CDK-cyclin part) between Capsaspora, two yeasts (Saccharomyces cerevisiae and Schizosaccharomyces pombe) and humans. The manuscript is clearly written and the results are well described. I am not an expert in the molecular biology of the cell cycle, but from an outsider's perspective it appears that the ability to obtain synchronized Capsaspora cell cultures and study the molecular biology of the cell cycle seems a considerable step toward studying the evolution of the cell cycle in animals and their relatives, but also more broadly in eukaryotes.

My main issue with the manuscript is the simplistic evolutionary comparisons and interpretations. I simply don't buy on the basis of the evidence presented the authors' evolutionary inference that the Capsaspora cyclin-CDK system "could represent an intermediate state in the evolution of animal-like cyclin-CDK regulation". Although that may end up being the case, my issue is that the authors' analyses haven't shown this. This is so because all their inferences are based on comparisons between Capsaspora and humans. But as the authors well know, there are many more animals than humans (what about Drosophila, non-vertebrate chordates or non-bilaterians?) and many more protists than Capsaspora (choanoflagellates, Ministeria, etc.). Similarly, there are many more outgroups than just fungi (the authors briefly mention Dictyostelium and plants) that would be very informative.

To address this issue, i believe the authors should greatly expand their search of CDK-cyclin genes in a broader sample of representative eukaryotes and add a figure that summarizes the distribution of these genes in these organisms.

Minor comments:

L26: "are present in all eukaryotes" -> "are widely / highly conserved in eukaryotes"

L31: "in a more representative opisthokont" - what is a "representative"opisthokont is highly subjective

L42-43: "could represent an intermediate state" - how can you distinguish this hypothesis from secondary simplification in Capsaspora?

L55: "yeasts" -> "fungi"

L140-141: This sentence should be stated as a hypothesis and not as an inference

L229-231: Can you rephrase? not sure i understand what you mean here

L326-327: how do you find this hypothesis to be "more parsimonious"?

L341: "ancient" -> "conserved" (the genes present in the Capsaspora genome are modern but conserved, they are not ancient)

L465-467: Which study was this? Please include a citation. And how much more limited was their sampling? All you've added is Capsaspora, which may or may not turn out to be a good model

Figure 7: i would definitely add more representative eukaryotic species here

Reviewer #3: The manuscript titled “Gradual evolution of cell cycle regulation by cyclin-dependent kinases during the transition to animal multicellularity” provides useful and important insights into the diversity and evolution of cell cycle control within the opisthokonts. The strength of the work lies in the development of a clean and reversible cell cycle synchronization protocol for vegitative Capsaspora cultures. The authors have capitalized on this achievement to identify cyclically regulated genes. They then go on to repurpose their analysis pipeline to identify cyclically expressed genes using published yeast, and both human Arabidopsis cell culture experiments to explore overlap in genes that are cyclically expressed.

Although the work presented in this manuscript is correlative, the experiments and analyses are appropriate first steps given how little is known about the biology of the system. Obvious next steps will be to prove that cyclical expression is dependent on transcription, identify transcriptional regulators (the transcription factor binding sites and proteins they bind), and/or show that the proposed regulators are actually controlling the cell cycle. Although it is tempting to ask for these analyses in the present paper, each item in the above list would require months (or years) of work to in a non-model system, are unnecessary to support the core claims made by the authors, and would drastically alter the scope of the paper and therefore necessitate multiple rounds of peer review.

The following points, however, should be addressed to ensure that the claims made in the paper are supported by the provided evidence, as well as to enhance the value of the reported data through discussion and comparison of existing literature.

Major points:

Major Point 1: In my opinion, the title is unnecessarily hyperbolic, and echos the oversimplification depicted in Figure 7 that depicts Capsaspora as an intermediate between human cells and yeast (which has been, as the authors describe, shown to be divergent relative to other fungi and protists). The data in the paper does indicate that aspects of human cell cycle regulation were already in place before the last common ancestor of animals. However, to extend that finding to claim that there was a gradual accumulation of cell cycle complexity associated with the evolution of animal multicellularity requires analysis of multiple animal lineages (Drosophila might be one to look at, among others) and additional non-opisthokonts are needed (Dictyostelium comes to mind). This may be possible to do by adding a paragraph discussion section that compares the new analyses to previous analyses of cell cycle regulation in additional lineages. If this is not feasible, the text and title should be edited to align with the findings of the paper.

Major Point 2: The sequencing and cytometry results are based on two replicate experiments. Given the number of timepoints involved, 3 biological replicates (as is customary in the field) may not have been feasible. However, this makes it even more important to describe in detail how distinct the two replicates are. The methods as described beginning on Line 522 suggest a scenario wherein a culture was split into 2 flasks, and were later treated treated in parallel. If this is correct, it should be stated as such. Furthermore, it may be more accurate to call these technical replicates as sister flasks may not display the biological variability needed for biological replicates (biological replicates should, if at all possible, be separated in time to ensure robust data).

Major Point 3: The decision to select genes for further analysis by ranking of P-values and then selection of a specific number genes should be justified (in contrast to analyzing all genes that meet a certain threshold p-value). For each analysis where this approach was taken, the maximum p-values included for the analysis should be reported, including the appropriate multiple-testing correction.

Major Point 4: Because this organism is diverged from model organisms, the behavior of Capsaspora cells after release from HU should be compared to what happens in other organisms. Is the 2 hour lag from wash out of HU typical (normalizing for total cell cycle time)? If not, what might be going on?

Major Point 5: The method section for synchronization does not have enough detail to replicate the experiments. How many times were cells washed? With what volume of media? It may be useful to others in this and related fields to include a step-by-step protocol as a supplemental document.

Minor Points:

Minor Point 1: In the abstract, it is stated that Capsaspora is “more representative” of opisthokonts. Without a large sample size across a large diversity of lineages, it is impossible to say whether a particular phenotype or genotype is “more representative”. Most likely, the authors simply mean that we need to broaden our perspective (which is clearly true and an important point) and can clarify the language here.

Minor Point 2: In the initial description of cell cycle synchronization (paragraph that begins on Line 159), it would be useful to state the doubling time of untreated adherent Capsaspora cells. This would provide context with which to evaluate the timing of cell cycle synchronization with hydroxyurea. If this number is unknown, it should be calculated and reported here.

Minor Point 3: In line 198, it states that it has been previously reported that Capsaspora has no gamma-tubulin ring complex, and cites a review on the evolution of MTOCs. I was not able to find reference to a lack of gamma-tubulin ring complexes in Capsaspora in the indicated paper (Figure 1 of the cited paper indicates Capsaspora does not have a SPD-2/CEP192 homolog, but this is not part of the gamma tubulin ring complex). Moreover, lack of Gamma tubulin and GCP2, 3, 4, 5, and 6 would be very surprising as this complex typically is used to nucleate microtubules in species whether or not they have MTOCs. But, more to the point, the presence or absence of the gamma-turc complex is a separate question from whether or not cells display centrosomes. The authors should clarify exactly what they mean here and ensure that the citations are what they intend.

Minor Point 4: The manuscript would benefit from copy editing for clarity. In particular, the sentence beginning on Line 344 is very confusing.

Minor Point 5: The cells in the panels of Fig. 2A are too small. Over half of each image could be cropped, and the remaining portion enlarged to allow the reader to see the dividing cell. Also, a time point before the cells round up should be included. Finally, the most common term is alpha-tubulin (not tubulin-alpha).

Minor Point 6: Fig. 2C axis units should be %age of cells in mitosis (the methods indicate that 100 cells were analyzed per time point).

Minor Point 7: Fig. 6: A, right panel AND Fig 6 B: These charts are not intuitive, nor are they well described in the figure legend. These panel need to be redesigned to be understandable and/or the figure legend written to walk the reader step-by-step through what is contained in the panel.

Minor Point 8: Fig 6C: the three colors are too close in hue to be easily discerned.

Reviewer #4: PGENETICS-D-19-01130 Perez-Posada et al.

The authors present a beautiful cell-cycle analysis of the unicellular, non-model organism Capsaspora owczarzaki, a species related to animal cells. They developed a synchrony/release time-series protocol using HU as a synchronization agent and analyzed cell-cycle progression by microscopy, flow cytometry, and RNA-seq. The punchline of the study from the abstract, summary and text appears to be that C. owczarzaki is likely an intermediate between yeasts and animal cells where orthologs of cyclins A, E, and B are binding to a single CDK. As well, the authors suggest that the cell cycle transcriptional program in C. owczarzaki is more similar to humans than yeasts.

The data presented are beautiful and are a very welcome addition to the burgeoning collection of high quality cell-cycle-time-series data in the community. However, a punchline that C. owczarzaki is an intermediate species with cyclin A, B, and E orthologs but only one CDK is a bit confusing, as that evidence comes largely from genome sequence data, and not the transcriptome (or microscopy/cytometry) data presented. Expansion of the CDK genes in animal cell lineages could also be observed directly from genome sequence. Thus, the major conclusion stemming from the transcriptome data analysis seems to be that the cell-cycle transcriptional program is more similar to humans than yeast. Comparing these programs is an important and difficult task as the underlying gene regulatory networks controlling the program in yeast has been suggested to function as the central cell-cycle oscillator. As presented, the conclusion that the program is more similar to humans is not obvious and needs to be specified further. As well, it is not clear that the conclusion is robust to the analytical methods, so some sensitivity testing should be done.

Specific comments:

1) On line 422 the authors state; “Overall, our cross-species comparison of the periodic gene expression programs revealed that the Capsaspora periodic gene expression program is more similar to human cells that to current unicellular model systems for the cell cycle…”. It is not clear what the authors mean by “more similar”. What precisely is the similarity metric they are using? Number of orthologs? Number of periodic orthologs? Ordering of orthologous genes? Number of genes in each cluster?

There are numerous comparisons one could make, especially in higher dimensions. These metrics need to be specified. Any metric that involves making calls on orthologs or making calls on periodic gene lists may be sensitive to the choices made by the methods. Thus, care needs to be made to test the sensitivity of these calls (See below).

The authors reference the work of Buchler, Skotheim, and Cross, who discovered that although the sequences of the genes making up the gene regulatory network controlling cell cycle transcription are unrelated in yeast and humans, the network topology is identical. This finding should be discussed as it suggests an additional way to measure evolutionary similarity beyond sequence homology.

2) There is no clear or obvious way to define cell-cycle regulated genes. There are many algorithmic methods that produce rank-ordered lists using a variety of definitions of “periodic”. The tricky part is always setting a sensible cut-off because these lists tend not to be bimodal (cell-cycle-regulated and not cell-cycle-regulated). The authors set a cut-off for their comparisons at 10% of the total genome, and it seems like this could influence their measure of periodic orthology. For the S. cerevisiae, the estimates of multiple studies suggest that ~ 20% of the genome appears to be under substantial cell cycle control. It makes sense that a single cell organism may have a greater part of its genome dedicated to cell division. The authors should broaden that cut-off and once the define similarity metrics precisely, they can ask whether their conclusions are robust to the choice of the size of the periodic gene sets they are querying.

3) Although there is no right or wrong way to define a periodic gene set, the method used by the authors is somewhat confusing. JTK cycle requires the user to input an expected period length or range of period lengths. Although I have no experience with RAIN, it appears it has a similar requirement. In Figures 1, 3, and 4 the authors make the case for the fact that in the 13.25-hour time-series experiment, they see cells return to the initial G1 (G1/S) state (By the way, the flow cytometry data says 13.25 in figure1 and Figure 4a the RNA-seq data says 13.15). The return to G1 can be observed in the flow cytometry data (Fig. 1) and the RNA-seq data (Fig. 4), arguing the cell-cycle period is in the 13-hour range. Yet, in the JTK analysis, the authors used a range of periods from 14 to 16 hours, and for the RAIN algorithm, they used a period of 16 hours. The authors need to justify these choices as they do not fit the data presented. Again, they should demonstrate that their choice of periods does not influence their final conclusions.

4) I don’t have the expertise to evaluate their methods for making calls on orthology, but it also seems like a complex and tricky business that could substantially influence the conclusions.

5) There is growing evidence that evolutionary distance or mechanism can be determined simply by orthology is short-sighted. Given the findings of Buchler, Skotheim, and Cross on the evolution of cell cycle transcription in humans and fungal lineages, it would be nice to see the work presented here in a broader context. Are there E2F/SBF/MBF and Rb/Whi5 orthologs? Any evidence they are controlling G1/S transcription or acting in similar network motifs?

Small comments:

On line 225, the authors state; “ we confirmed that top-ranked genes showed oscillatory behavior”. How exactly was oscillatory behavior confirmed? I’m guessing it was visual inspection? The eyeball test is certainly a legitimate method for checking, but whatever the method is should be specified.

For Figure 4A, the authors show normalized expression values. From the description in the methods I think what they are presenting is a Z-score. If it is, it should be labeled as such in the figure. If it’s not, then some rationale for the normalization method should be stated.

**Have all data underlying the figures and results presented in the manuscript been provided?**

Reviewer #1: Yes

Reviewer #2: Yes

Reviewer #3: Yes

Reviewer #4: Yes

PLOS authors have the option to publish the peer review history of their article (what does this mean?). If published, this will include your full peer review and any attached files.

Reviewer #1: No

Reviewer #2: No

Reviewer #3: No

Reviewer #4: No

---

## [Decision Letter · Decision Letter 1]

23 Dec 2019

Dear Iñaki,

We are pleased to inform you that your revised PLOS Genetics manuscript entitled "Cell cycle transcriptomics of Capsaspora provides insights into the evolution of cyclin-CDK machinery" has been editorially accepted for publication in PLOS Genetics. Congratulations!  The manuscript was reviewed again by the same three reviewers who reviewed the original version, and all three found the revised version to be acceptable as submitted with no further revisions.  We very much appreciate your careful attention to detail and responsiveness in the revised manuscript. All three reviewers also commented on the importance of the work, and the quality.

Thank you again for supporting open-access publishing; we are looking forward to publishing your work in PLOS Genetics!  We thank you for entrusting your best work with us, and look forward to future submissions from you as your research program in this exciting field continues to advance.

All best wishes for the holidays!

Yours sincerely,

Joe

Joseph Heitman, MD, PhD

Associate Editor

PLOS Genetics

Bret Payseur

Section Editor: Evolution

PLOS Genetics

Comments from the reviewers (if applicable):

Reviewer's Responses to Questions

**Comments to the Authors**:

Reviewer #1: Great revision. Authors addressed my critiques with extra experiments, analysis, and rewriting sections of the manuscript.

Reviewer #2: The authors have improved the manuscript and toned down the evolutionary speculation, which was my major concern.

Reviewer #3: In this revised version of their manuscript, the authors have addressed all of the major and minor points I had raised about the first version of the paper, and also addressed most points raised by other referees. The result is an interesting contribution from an emerging model system with insights supported by the evidence and methods that should allow for the work to be built up on by other laboratories.

**Have all data underlying the figures and results presented in the manuscript been provided?**

Reviewer #1: Yes

Reviewer #2: Yes

Reviewer #3: Yes

PLOS authors have the option to publish the peer review history of their article (what does this mean?). If published, this will include your full peer review and any attached files.

Reviewer #1: No

Reviewer #2: No

Reviewer #3: No

**Data Deposition**

http://datadryad.org/submit?journalID=pgenetics&manu=PGENETICS-D-19-01130R1

**Press Queries**

---

## [Editor Report · Acceptance letter]

4 Mar 2020

PGENETICS-D-19-01130R1 

Cell cycle transcriptomics of Capsaspora provides insights into the evolution of cyclin-CDK machinery 

Dear Dr Ruiz-Trillo, 

We are pleased to inform you that your manuscript entitled "Cell cycle transcriptomics of Capsaspora provides insights into the evolution of cyclin-CDK machinery" has been formally accepted for publication in PLOS Genetics! Your manuscript is now with our production department and you will be notified of the publication date in due course.

With kind regards,

Kaitlin Butler

PLOS Genetics

On behalf of:
